# ATTRIBUTED GRAPH CLUSTERING VIA MODULARITY AIDED COARSENING

## ABSTRACT

Graph clustering is a widely used technique for partitioning graphs, community detection, and other tasks. Recent graph clustering algorithms depend on combinations of the features and adjacency matrix, or solely on the adjacency matrix. However, in order to achieve high-quality clustering, it is necessary to consider all these components. In this paper, we propose a novel unsupervised learning framework that incorporates modularity with graph coarsening techniques and important graph regularization terms that improve the clustering performance. Furthermore, we also take into account Dirichlet energies for smoothness of signals, spectral similarity, and coarsening reconstructional error. The proposed framework is solved efficiently by leveraging block majorization-minimization, $\log \det$ of the Laplacian, smoothness and modularity, and is readily integrable with deep learning architectures such as GCNs and VGAEs in the form of losses. Extensive theoretical analysis and experiments with benchmark datasets elucidate the proposed framework's efficacy in graph clustering over existing state-of-the-art methods on both attributed and non-attributed graphs.

## 1 INTRODUCTION

Graph clustering partitions the nodes of a graph into groups based on their specific attributes and interaction patterns learnt from the adjacency matrix and the features. The major applications of graph clustering are social network analysis (Tang et al., 2008), genetics and bio-medicine (Cheng & Ma, 2022) (Buterez et al., 2021), knowledge graphs (Hamaguchi et al., 2017), and computer vision (Mondal et al., 2021) (Caron et al., 2018), etc. The recent surge in demand for unsupervised graph learning techniques because of a surge in data collection has made it infeasible to label the majority of datasets manually. Researchers are bound to resort to unsupervised methods of analysis.

Cut-based and Modularity metrics are the two categories of measures used to determine the quality of partitioning of a graph. Methods optimizing cut-based metrics are usually based on the fact that the Fiedler vector (the eigenvector of the second smallest eigenvalue of the Laplacian) produces a graph cut minimal in the number of edges (Fiedler, 1973; Newman, 2006). However, Leskovec et al. (2008) show that this does not translate well for real-world graphs as the partitions formed do not need to balanced in size (nodes) and cluster quality decreases with an increase in cluster size. Some methods try to normalize these partitions, such as Ratio Cut (Wei & Cheng, 1989) which considers the number of nodes and Normalized Cut (Shi & Malik, 2000) which considers total edge volume of the partition. MinCutPool (Bianchi et al., 2020) performs graph pooling using normalized cuts as a regularizer. A major disadvantage of cut-based metrics is that they assume each node gets mapped to one cluster, which is not consistent with real graph data.

Modularity has been popularized because it measures how different the edge density of a graph is compared to a random graph of the same degree sequence. This can also be viewed as a statistical approach incorporating a null model. A significant edge that modularity offers is that its definition can be extended to directed graphs with overlapping clusters (Nicosia et al., 2009), making it suitable for complex networks. In this paper, we study graph clustering through the lens of modularity. However, it too has drawbacks on small clusters such as its resolution limit (Fortunato & Barthélemy, 2007).

Graph coarsening condenses a large graph into a smaller one while preserving its properties. Feature Graph Coarsening (FGC) (Kumar et al., 2023) is the first optimization framework that takes into account node features instead of just the adjacency. More about this is discussed in Section 2. Recent Graph Neural Network (GNN)-based approaches utilize both node features and graph topology. Most state-of-the-art clustering methods use Variational Graph Auto-Encoder (VGAE)-based

backbone because they offer superior performance and have flexibility to use different task-specific encoders/decoders. The existing GNN-free approaches, on the other hand, take only graph topology into account. This is disadvantageous because many real-word graphs have important/meaningful node features and satisfy important homophily and smoothness assumptions, i.e., if two nodes are connected, they should be similar.

**Motivation.** We aim to utilize the feature graph coarsening framework (which does not perform well on clustering) for clustering. With this in mind, we add a modularity maximization term because it is a good measure for community detection. This is discussed in 2. This comprises the Q-FGC method and greatly improves clustering performance. We also utilize various GNN architectures and integrate our loss function, resulting in Q-GCN, Q-VGAE, and Q-GMM-VGAE and outperform other methods. An important future work can entail utilizing contrastive learning in our framework to get the best of both worlds.

**Key Contributions:**

- We introduce a new approach to clustering by introducing the first optimization-based framework for attributed graph clustering via coarsening using modularity. Our approach is highly efficient and theoretically convergent. We also reflect on how our formulation overcomes the disadvantages of existing methods.

- We show that the proposed clustering objective can be easily integrated with GNN-based architectures. This allows us to leverage the message-passing mechanism of GNNs to further enhance the performance of our method.

- We conduct extensive experimental validation on real-world and synthetic datasets, showing that our method outperforms existing state-of-the-art methods.

We show a new approach to clustering, as most methods don't take into account various important terms such as the Dirichlet energy (smoothness) of the coarsened graph, modularity maximization, a term to ensure the coarsened graph is connected, and a regularizer for balanced mapping of nodes to clusters, preventing collapse. Our method can also directly observe the relationship between clusters, and also represent each cluster. This is quite important for performing first-hand analysis on large unlabelled datasets.

**Novelty** This may seem like an incremental work because our modification to the FGC objective is one term. However, here are some points to consider:

- A significant increase in performance from baseline models FGC (**+153%**) and DMoN (**+40%**), which shows that our contribution is to **allow potentially any coarsening method to work in a clustering setting**, as there are *theoretical benefits* in coarsening literature that have not been studied in clustering yet.

- Comprehensive analyses with theoretical guarantees, including supporting proofs of convexity, proofs of KKT optimality, proofs of convergence, ablation studies on the behavior of the loss terms and how it differs from FGC, recovery on a DC-SBM, comparison of runtime and complexities, and also comparison of modularity to make it empirically and theoretically comprehensive.

- Experiments that are not limited to a few small datasets but range from benchmark, to non-attributed, to very large datasets, even though our method does not specialize for them.

## 2 RELATED WORKS

The focus is on three types of existing graph clustering approaches: (1) Graph coarsening methods, (2) VGAE-based clustering (Section 3), and (3) Spectral modularity maximization techniques.

**Graph Coarsening and Pooling Methods.** DiffPool (Ying et al., 2018)learns soft cluster assignments at each layer of the GNN and two extra losses, entropy to penalize the soft assignments and a link prediction based loss. SAGPool (Lee et al., 2019) calculates attention scores and node embeddings to determine the nodes that need to be preserved or removed. Top-k (Gao & Ji, 2019) also works by sparsifying the graph with the learned weights. MinCutPool (Bianchi et al., 2020) formulates a differentiable relaxation of spectral clustering via pooling. However, Tsitsulin et al. (2023) show that it's orthogonal regularization dominates over the clustering objective and the objective is not optimized. Some disadvantages of these methods are stability and computational complexity in the case of SAGPool and DiffPool and convergence in MinCutPool.

**Modularity Optimization.** In theory, a higher value of modularity (**Q**) is associated with better quality clusters. However, maximizing modularity over all partitions of a graph is **NP-hard** (Brandes et al., 2008). Various heuristic algorithms have been established that solve this problem including sampling, simulated annealing (sim, 2005) (Guimerà & Amaral, 2005), mathematical programming (Agarwal, G. & Kempe, D., 2008), and greedy agglomerative algorithms (Newman, 2004; Blondel et al., 2008a). The usage of these algorithms has plummeted over the years because they rely solely on the topological information of graphs and ignore node features. Another major drawback is that they require intensive computation and are thus impractical for large-scale networks. The Louvain (Blondel et al., 2008b) and Leiden algorithms improve that. With the development of GNNs, this also improved drastically; however, modularity maximization using GNNs still needs to be studied. To the best of our knowledge, Deep Modularity Network (DMoN) (Tsitsulin et al., 2023) and Modularity-Aware GAE (Salha-Galvan et al., 2022) are the only deep learning architectures to use modularity in training and are crucial baselines for our method. DMoN's clustering objective optimizes only for modularity (with a collapse regularization to prevent all nodes being assigned one cluster) but does not consider smoothness of signals and offers no theoretical guarantees about convergence. Modularity-Aware GAEs and VGAEs use a prior membership matrix using Louvain algorithm and optimize for modularity using RBF kernel as a same-community assignment proxy.

**Deep Graph Clustering.** Previous literature can be classified based on contrastive and non-contrastive methods. On the non-contrastive side, Pan et al. (2018) proposed ARGA and ARVGA, enforcing the latent representations to align to a prior using adversarial learning. By utilizing an attention-based graph encoder and a clustering alignment loss, Wang et al. (2019) propose DAEGC. Subsequently, SDCN (Bo et al., 2020) propose to learn a "structure-aware" representation by their delivery operator. Liu et al. (2022) design the DCRN model to alleviate representation collapse by a propagation regularization term minimizing the JSD between the latent and its product with normalized $A$. Contrastive methods include AGE (Cui et al., 2020) which builds a training set by adaptively selecting node pairs that are highly similar or dissimilar after filtering out high-frequency noises using Laplacian smoothing. Zhao et al. (2021) propose GDCL to correct the sampling bias by choosing negative samples based on the clustering label. Liu et al. (2023a) propose SCGC, which uses two MLPs to get augmented node features, and then find a cross-view similarity matrix to contrast against $A$.

## 3 BACKGROUND

In this section, we introduce the concepts that play a foundational role in the formulation of our method.

**Notations.** Let $G = \{V, E, A, X\}$ be a graph with node set $V = \{v_1, v_2, ..., v_p\}$ ($|V| = p$), edge set $E \subset V \times V\}(|E| = e)$, weight (adjacency) matrix $A$ and node feature matrix $X \in \mathbb{R}^{p \times n}$. Also, let $\mathbf{d} \in \mathbb{Z}_+^p$ be the degree vector. Then, the graph Laplacian is $\Theta = \text{diag}(\mathbf{d}) - A$ and the set of all valid Laplacian matrices is defined as: $S_\Theta = \{\Theta \in \mathbb{R}^{p \times p} | \Theta_{ij} = \Theta ji \leq 0 \text{ for } i \neq j, \Theta_{ii} = \sum_{j=1}^p \Theta_{ij}\}$

### 3.1 GRAPH COARSENING

Graph coarsening is a classical method used in large-scale machine learning to construct a smaller graph $G_c$ from the original graph $G = \{V, E, A, X\}$ while preserving properties of $G$. The commonly used measures of similarity are hyperbolic error (Bravo Hermsdorff & Gunderson, 2019), reconstruction error, $\epsilon$- similarity (Loukas, 2019), and spectral similarity. For the coarsened graph $G_c$, we denote the new vertex set as $\tilde{V}(|\tilde{V}| = k)$ features as $\tilde{X} \in \mathbb{R}^{k \times n}$ and the Laplacian $\Theta_C$. We define $C \in [0, 1]^{p \times k}$ to be the *soft* cluster assignment matrix (i.e. each non-zero entry of matrix C i.e., $C_{ij}$ indicates probability of $i$-th node of $G$ mapped to $j$-th cluster or supernode of $G_c$). The coarsening matrix $C$ plays the role of cluster assignment matrix in our study, i.e. $k$ represents the number of clusters. Moreover, the Laplacian and feature matrix of the coarsened graph and original graph together satisfy the following properties:

$$X = C\tilde{X}, \quad \Theta_C = C^T \Theta C \text{ where, } C \in \mathcal{S}_c \tag{1}$$

$$\mathcal{S}_c = \{C \in \mathbb{R}_+^{p \times k} | \langle C_i, C_j \rangle = 0 \ \forall \ i \neq j, \langle C_l, C_l \rangle = d_i, ||C_i||_0 \geq 1 \text{ and } ||[C^T]_i||_0 = 1\} \tag{2}$$

Kumar et al. (2023) proposed the first optimization-based framework FGC to incorporate both graph topology using the Laplacian matrix and node features. Their method preserves important qualities of the original graph $G$ in the coarsened graph $G_c$.

## 3.2 Variational Graph Auto Encoders (VGAEs)

VGAEs (Kipf & Welling, 2016a) are an increasingly popular class of GNNs that leverage variational inference techniques for learning latent representations for graphs in unsupervised settings. A typical architecture involves using GCN-based encoders to transform a high-dimensional graph to a low-dimensional space, followed by a decoder to reconstruct the adjacency matrix. Due to its ability to attain competitive performance on a multitude of tasks, including node classification and link prediction, it is the preferred backbone for contemporary graph-based architectures.

Many attempts have been made to use VGAEs with k-means on latent embeddings, but it has been unsuitable for clustering. This is primarily because embedded manifolds obtained from VGAEs are curved and must be flattened before any clustering algorithms using Euclidean distance are applied. Refer to supplementary material K for an explanation. VGAEs only use a single Gaussian prior for the latent space, whereas clustering requires the integration of meta-priors. Additionally, the inner-product decoder fails to capture locality and cluster information in the formed edges. Several clustering-oriented variants of VGAEs (Mrabah et al., 2022) (Hui et al., 2020) have been developed that overcome most of these challenges. GMM-VGAE (Hui et al., 2020) is one such architecture and is relevant to our paper. GMM-VGAE partitions the latent space using a Gaussian Mixture Model and assigns a separate prior for each cluster to better model complex data distributions instead of the single prior in VGAEs. Despite the improvement in performance, it's inner-product decoder still suffers from the same problems.

## 3.3 Spectral Modularity Maximization

Spectral Clustering is the most direct approach to graph clustering, where we minimize the volume of inter-cluster edges (i.e. the total number of edges in between clusters and not inside them). Modularity is the difference the number of edges between nodes in a cluster $\mathbf{C_i}$ and the expected number of such edges in a random graph with identical degree sequence. It is mathematically defined as

$$\mathcal{Q} = \frac{1}{2e} \sum_{i,j} \left[ A_{ij} - \frac{d_i d_j}{2e} \right] \delta(c_i, c_j) \tag{3}$$

where $\delta(c_i, c_j)$ is the Kronecker delta. Maximizing this form of modularity is NP-hard (Brandes et al., 2008). However, we can approximate it with a spectral relaxation, which involves a modularity matrix $B$. The modularity matrix $B$ and spectral modularity are defined as:

$$B = A - \frac{\mathbf{dd}^T}{2e}, \quad \mathcal{Q} = \frac{1}{2e} Tr(C^T B C) \tag{4}$$

Similar to the Graph Laplacian matrix $\mathbf{\Theta}$, $\mathbf{B}$ is symmetric and is defined to have a row-sum and column-sum as 0, thereby making $\mathbf{1}$ as one of it's eigenvector and 0 as the corresponding eigenvalue. These spectral properties of the modularity matrix are also seen in the Laplacian, as noted in Newman (2006), which is of course a crucial element in spectral clustering. Also, modularity is maximized when $u_1^T s$ is maximized where $u$ are eigenvectors of $B$ and $s$ is the community assignment vector, i.e., placing the majority of the summation in $Q$ on the first (and largest) eigenvalue of $B$. Moreover, modularity is closely associated with community detection. These special spectral properties make $\mathbf{B}$ as the ideal choice for graph clustering.

## 4 Proposed Method

In this section, we present our proposed clustering framework that overcomes the aforementioned drawbacks of existing methods and achieves competitive performance on six benchmark datasets.

### 4.1 Modularity aided Feature Graph Coarsening

Following the success of FGC, we pose the clustering problem as a special case of graph coarsening, when the number of nodes in the coarsened graph is equal to the number of clusters. FGC performs excellently for coarsening ratios $(k/p)$ in the range of 0.01-0.1, preserving the eigenvalue distribution of the original graph. However, for clustering purposes, we usually have $p$ of the order of thousands, whereas $k$ is usually less than 10, implying a coarsening ratio lower than 0.001. So empirically and experimentally (Table 2), it is evident that FGC alone is inadequate. We assume that the original graph is smooth considering that most real world graphs are smooth and homophilic. Note that this doesn't mean our method fails on synthetic graphs. We have studied this in the ablation study 5.5, and is able to recover the community structure completely. We introduce an optimization-based framework for attributed graph clustering via coarsening with modularity as follows:

$$\min_{\tilde{X},C} \mathcal{L}_{MAGC} = \text{tr}(\tilde{X}^T C^T \Theta C \tilde{X}) + \frac{\alpha}{2} \left\| C\tilde{X} - X \right\|_F^2 - \frac{\beta}{2e} \text{tr}(C^T BC) \tag{5}$$

$$- \gamma \log \det(C^T \Theta C + J) + \frac{\lambda}{2} \left\| C^T \right\|_{1,2}^2$$

$$\text{subject to } \mathcal{C} = \{C \in \mathbb{R}^{p \times k} | C \geq 0, \left\| C_i^T \right\|_2^2 \leq 1\} \, \forall i \text{ where, } J = \frac{1}{k} \mathbf{1}_{k \times k}$$

$$\tag{6}$$

The significance of each term is term in the optimization objective is:
- $\text{tr}(\tilde{X}^T C^T \Theta C \tilde{X})$ is the Dirichlet energy of the coarsened graph (because $\Theta_C = C^T \Theta C$), a measure of the smoothness of it's signals. Graph-based modeling is based on the assumption that graph signal variations are smooth between connected nodes. This term ensures the smoothness property of the original graph is passed onto the coarsened one.

- $\left\| C\tilde{X} - X \right\|_F^2$ simply enforces the relaxation of the constraint $X = C\tilde{X}$, i.e., the large graph $X$ gets coarsened to the the smaller $\tilde{X}$ with the cluster assignment/loading matrix $C$.

- $\text{tr}(C^T BC)$ is the modularity term for better cluster quality. The negative of this is added to the loss as we want to maximize modularity while minimizing the loss. This is discussed in 3.3

- $-\log \det(C^T \Theta C + J)$ is the log-determinant term to ensure the coarsened graph is connected. This works because it can be written as $-\sum_i \log \lambda_i$ where $\lambda_i$'s are the eigenvalues of the $\Theta_C$. By minimizing this, we are ensuring that minimal number of $\lambda_i$'s are 0, since the number of connected components in a graph is equal to the multiplicity of 0 in it's laplacian eigenvalues.

- $\left\| C^T \right\|_{1,2}^2$ is an $\ell_{1,2}^2$ norm regularizer for a balanced mapping of nodes to clusters (i.e., every cluster has at least one node and each node is mapped to a cluster) (Kim & Park, 2007).

**Update rules.** Since the resulting optimization problem is multi-block non-convex and there is no closed-form solution, we use Block Majorization-Minimization framework similar to FGC (Kumar et al., 2023). We iteratively solve the problem by updating $\mathbf{C}$ and $\tilde{\mathbf{X}}$ alternatively while keeping the other constant. These iterations are performed until convergence or until some stopping criteria is met.

**Lemma 1.** *The problem equation 6 with respect to $\tilde{X}$ while keeping $C$ constant is a convex optimization problem.*

*Proof.* As $\text{tr}(\tilde{X}^T C^T \Theta C \tilde{X})$ and $\| C\tilde{X} - X \|_F^2$ are convex functions with respect to $\tilde{X}$ while keeping $C$ constant and $\tilde{X} \in \mathbb{R}^{p \times n}$ together makes the problem equation 6 with respect to $\tilde{X}$ as a convex optimization problem. $\square$

**Lemma 2.** *The problem equation 6 with respect to $C$ is convex optimization problem while keeping $\tilde{X}$ constant.*

*Proof.* All the terms in the objective function of problem 6 with respect to $C$ while keeping $\tilde{X}$ are convex functions and the set $\mathcal{C}$ is a closed convex set together makes the problem a convex optimization problem. More details are in supplementary material. $\square$

Considering the objective function with respect to $C$ as $f(C)$, The majorized function of $f(C)$ at $C^t$ using the first order taylor series expansion is:

$$g(C|C^t) = f(C^t) + (C - C^t)\nabla f(C^t) + \frac{L}{2} \left\| C - C^t \right\|^2 \tag{7}$$

$$\min_{C \in \mathcal{S}_C} \frac{1}{2} C^T C - C^T \left( C^t - \frac{1}{L} \nabla f(C^t) \right) \tag{8}$$

Equation 8 is the majorized problem of equation 6. The optimal solution to Eqn. 8, found by using Karush–Kuhn–Tucker (KKT) optimality conditions is (Proof is deferred to the supplementary material B):

$$C^{t+1} = \left( C^t - \frac{1}{L} \nabla f(C^t) \right)^+ \tag{9}$$

$$\text{where, } \nabla f(C^t) = -2\gamma \Theta C^t (C^{t^T} \Theta C^t + J)^{-1} + \alpha (C^t \tilde{X} - X)\tilde{X}^T + 2\Theta C^t \tilde{X} \tilde{X}^T + \lambda C^t \mathbf{1}_{k \times k}$$

$$- \frac{\beta}{e} BC^t \tag{10}$$

$$\tilde{X}^{t+1} = \left( \frac{2}{\alpha} C^T \Theta C + C^T C \right)^{-1} C^T X \tag{11}$$

**Convergence Analysis.**

**Theorem 1.** *The sequence $\{C^{t+1}, \tilde{X}^{t+1}\}$ generated by Algorithm 1 converges to the set of Karush–Kuhn–Tucker (KKT) optimality points for Problem 6*

*Proof.* The detailed proof can be found in the supplementary material C.  □

**Complexity Analysis.** The worst-case time complexity of a loop in Algorithm 1 is $\mathcal{O}(p^2 k + pkn)$ because of the matrix multiplication of $\Theta(p \times p)$ with $\mathbf{C}(p \times k)$ and matrix multiplication of $\mathbf{C}(p \times k)$ with $\tilde{\mathbf{X}}(k \times n)$ in the update rule of $\mathbf{C}$ equation 9. Note that in clustering, $k$ is much smaller than $p$. This makes our method much faster than previous optimization based methods and faster than GCN-based clustering methods which have complexities around $\mathcal{O}(p^2 n + pn^2)$. We discuss this more in Supplementary Material L.

---

**Algorithm 1** Q-FGC Algorithm

---

**Require:** $G(\mathbf{X}, \Theta), \alpha, \beta, \gamma, \lambda$
1: $t \leftarrow 0$
2: **while** Stopping Criteria not met **do**
3:     Update $\mathbf{C}^{t+1}$ as in Eqn. 9 and Update $\tilde{\mathbf{X}}^{t+1}$ as in Eqn. 11
4:     $t \leftarrow t + 1$
5: **end while**
6: **return** $\mathbf{C}^t, \tilde{\mathbf{X}}^t$

---

### 4.2 INTEGRATING WITH GNNs

Our optimization framework is easily integrable with deep learning methods (GNNs) by adding equation 6 in the form of a loss function to be minimized using gradient descent. We show this integration and their results on the most widely-used architectures like Graph Convolutional Networks (GCNs) (Kipf & Welling, 2016b), VGAEs(Kipf & Welling, 2016a), and a variant of VGAE (GMM-VGAE) (Hui et al., 2020). The former two are popular because of the relative simplicity.

Note that for clustering, learning $C$ is of more importance to us than learning $\tilde{X}$, as it tells us about which nodes belong to which clusters, whereas $\tilde{X}$ tells us about a mean/prototypical element in that cluster, which can't be compared directly with the ground truth labels. So, we just learn $C$ instead of $\tilde{X}$, and calculate it in each iteration by multiplying $X$ with the pseudo-inverse of $C$ because of equation 1.

**Q-GCN.** We integrate our loss function ($\mathcal{L}_{MAGC}$ equation 6) into a simple three-layer GCN model. We learn the soft cluster assignments $\mathbf{C}$ by taking it to be the output of the final GCN layer. So, our input features go from $p \times n$, to $p \times k$. The architecture and loss can be seen in Figure 1.

**Q-VGAE.** VGAEs operate by reconstructing the adjacency matrix.

We have provided a theoretical summary of VGAE in supplementary material F. The loss can be written as

$$\mathcal{L}_{VGAE} = \lambda_{recon} \underbrace{\mathbb{E}_{q(\mathbf{Z}|\mathbf{X},\mathbf{A})}[\log p(\hat{\mathbf{A}}|\mathbf{Z})]}_{\text{Reconstruction Error}} - \lambda_{kl} \underbrace{\text{KL}[q(\mathbf{Z}|\mathbf{X},\mathbf{A}) \,||\, p(\mathbf{Z})]}_{\text{Kullback-Leibler divergence}} \tag{12}$$

where, $\mathbf{Z}$ represents the latent space of the VGAE.

On top of this architecture, we add a GCN layer taking $Z$ as input and predicting $C$. So, for a VGAE, we are minimizing the sum of three losses: the reconstruction loss, the KL-divergence loss and our loss. $\mathcal{L}_{Q-VGAE} = \mathcal{L}_{MACG} + \mathcal{L}_{VGAE}$

**Q-GMM-VGAE.** This variant of VGAE uses a Gaussian Mixture Model (GMM) on the latent space to more effectively discover data distributions. This works well because it minimizes the evidence lower bound (ELBO/variational lower bound (Hui et al., 2020; Kingma & Welling, 2014; Kipf & Welling, 2016a)) using multiple priors instead of a single Gaussian prior in a normal VGAE. Intuitively, taking as many priors as number of clusters seems like a good idea, which is what Hui et al. (2020) do.

## 5 EXPERIMENTS

### 5.1 BENCHMARK DATASETS AND BASELINES

We evaluate our method on a range of datasets, ranging from small attributed datasets like Cora and CiteSeer to larger ones like PubMed, and even unattributed datasets like Airports (Brazil, Europe and USA). We also experiment with very large datasets like CoauthorCS/Physics, AmazonPhoto/PC and ogbn-arxiv. A summary of all the datasets used in our paper is given in the supplementary material D.

We compare the performance of our method against three types of existing state-of-the-art methods based on the provided input and type of architecture: a) methods that use only the node attributes; b) methods that only use graph-structure; c) methods that use both graph-structure and node attributes.

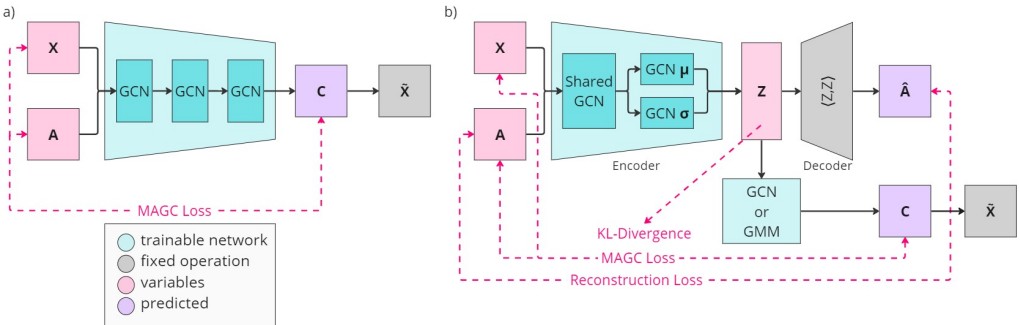

Figure 1: **a) The architecture of Q-GCN.** We train the encoder to learn the soft cluster assignment matrix $\mathbf{C}$. $\tilde{\mathbf{X}}$ is obtained using the properties of coarsened graphs $\tilde{\mathbf{X}}^{t+1} = \mathbf{C}^{t+1\dagger}\mathbf{X}$ (Eqn 1). Our proposed MAGC loss is now computed using $\mathbf{C}$ and $\tilde{\mathbf{X}}$.
**b) The architecture of Q-VGAE/Q-GMM-VGAE.** The three-layered GCN encoder, which takes $\mathbf{X}$ and $\mathbf{A}$ as input, learns the latent representation of the graph $\mathbf{Z}$. $\mathbf{Z}$ is then passed through an inner-product decoder to reconstruct the adjacency matrix $\hat{\mathbf{A}}$ Reconstruction loss is calculated between $\hat{\mathbf{A}}$ and $\mathbf{A}$ and KL-Divergence on $\mathbf{Z}$. In Q-VGAE (Q-GMM-VGAE), $\mathbf{Z}$ is parallelly passed through a GCN layer (GMM) to output the soft cluster assignments $\mathbf{C}$. MAGC loss is now computed in the same way as Q-GCN.

The last category can be further subdivided into three sets: i) graph coarsening methods; ii) GCN-based architectures; iii) VGAE-based architectures and contrastive methods; iv) largely modified VGAE architectures.

## 5.2 METRICS

We analyse the performance of our method using label alignment metrics which relate ground truth node labels to cluster assignments. We report Normalised Mutual Information (NMI), Adjusted Rand Index (ARI), and Accuracy (ACC). Higher value for these metrics are desirable. Please refer to Supplementary Material D for an explanation.

We chose NMI as the primary metric for selecting the most performant models because of these reasons and because it is considered most important by the majority of graph clustering literature.

**Training Details.** We have used the same architectures for Q-GCN, Q-VGAE and Q-GMM-VGAE throughout. Q-GCN is composed of 3 GCN layers with hidden sizes as 128 and 64. Q-VGAE and Q-GMM-VGAE are composed of 3 GCN layers for the encoder (1 shared with output size 128, and 1 each for $\mu$ and $\sigma$ of output size 64). Q-VGAE has an additional GCN layer after the latent space to generate $C$, whereas Q-GMM-VGAE utilises a GMM. More training details are available in the Supplementary Material E.

## 5.3 ATTRIBUTED GRAPH CLUSTERING

We highlight our key results on the three classical datasets Cora, CiteSeer, and PubMed (Sen et al., 2008) in Table 1. Our proposed method surpasses all existing methods in terms of NMI while also achieving competitive performance in terms of Accuracy and ARI. As mentioned above the best models were selected based on the NMI scores. The results for very large datasets are present in the supplementary material I. Note that we perform full-batch training (passing the whole graph) instead of randomly-sampled batches (which involves cutting the graph into multiple subgraphs, adding a great degree of bias as the community structure is broken) like in some other works such as S3GC (Devvrit et al., 2022). For very large graphs such as ogbn-arxiv, we have no choice but to use batching.

## 5.4 NON-ATTRIBUTED GRAPH CLUSTERING

For non-attributed graphs, we use a one-hot vector of the degree vector as the features. This is a primitive way of making features, and there are learning based methods such as DeepWalk (Perozzi et al., 2014b), node2vec (Grover & Leskovec, 2016) etc. This was done for the sake of fair comparison, as the other methods also use this as node features. We present our results in Table 2a. Clearly, we achieve competitive or even higher performance in terms of NMI even for non-attributed datasets.

| Method | Cora | | | CiteSeer | | | PubMed | | |
|---|---|---|---|---|---|---|---|---|---|
| | ACC ↑ | NMI ↑ | ARI ↑ | ACC ↑ | NMI ↑ | ARI ↑ | ACC ↑ | NMI ↑ | ARI ↑ |
| K-means | 34.7 | 16.7 | 25.4 | 38.5 | 17.1 | 30.5 | 57.3 | 29.1 | 57.4 |
| Spectral Clustering | 34.2 | 19.5 | 30.2 | 25.9 | 11.8 | 29.5 | 39.7 | 3.5 | 52.0 |
| DeepWalk (Perozzi et al., 2014a) | 46.7 | 31.8 | 38.1 | 36.2 | 9.7 | 26.7 | 61.9 | 16.7 | 47.1 |
| Louvain (Blondel et al., 2008b) | 52.4 | 42.7 | 24.0 | 49.9 | 24.7 | 9.2 | 30.4 | 20.0 | 10.3 |
| GAE [NeurIPS'16] (Kipf & Welling, 2016a) | 61.3 | 44.4 | 38.1 | 48.2 | 22.7 | 19.2 | 63.2 | 24.9 | 24.6 |
| DGI [ICLR'19] (Veličković et al., 2019) | 71.3 | 56.4 | 51.1 | 68.8 | 44.4 | 45.0 | 53.3 | 18.1 | 16.6 |
| GIC [PAKDD'21] (Mavromatis & Karypis, 2021) | 72.5 | 53.7 | 50.8 | 69.6 | 45.3 | 46.5 | 67.3 | 31.9 | 29.1 |
| DAEGC [IJCAI'19] (Wang et al., 2019) | 70.4 | 52.8 | 49.6 | 67.2 | 39.7 | 41.0 | 67.1 | 26.6 | 27.8 |
| GALA [ICCV'19] (Park et al., 2019) | 74.5 | 57.6 | 53.1 | 69.3 | 44.1 | 44.6 | 69.3 | 32.7 | 32.1 |
| AGE [KDD'20] (Cui et al., 2020) | 73.5 | 57.5 | 50.0 | 69.7 | 44.9 | 34.1 | 71.1 | 31.6 | 33.4 |
| DCRN [AAAI'22] (Liu et al., 2022) | 61.9 | 45.1 | 33.1 | 70.8 | 45.8 | 47.6 | 69.8 | 32.2 | 31.4 |
| FGC [JMLR'23] (Kumar et al., 2023) | 53.8 | 23.2 | 20.5 | 54.2 | 31.1 | 28.2 | 67.1 | 26.6 | 27.8 |
| **Q-FGC (Ours)** | 65.8 | 51.8 | 42.0 | 65.9 | 40.8 | 42.0 | 66.7 | 32.8 | 27.9 |
| **Q-GCN (Ours)** | 71.6 | 58.3 | 53.6 | 71.5 | 47.0 | 49.1 | 64.1 | 32.1 | 26.5 |
| SCGC [IEEE TNNLS'23] (Liu et al., 2023a) | 73.8 | 56.1 | 51.7 | 71.0 | 45.2 | 46.2 | - | - | - |
| MVGRL [ICML'20] (Hassani & Khasahmadi, 2020) | 73.2 | 56.2 | 51.9 | 68.1 | 43.2 | 43.4 | 69.3 | 34.4 | 32.3 |
| VGAE [NeurIPS'16] (Kipf & Welling, 2016a) | 64.7 | 43.4 | 37.5 | 51.9 | 24.9 | 23.8 | 69.6 | 28.6 | 31.7 |
| ARGA [IJCAI'18] (Pan et al., 2018) | 64.0 | 35.2 | 61.9 | 57.3 | 34.1 | 54.6 | 59.1 | 23.2 | 29.1 |
| ARVGA [IJCAI'18] (Pan et al., 2018) | 63.8 | 37.4 | 62.7 | 54.4 | 24.5 | 52.9 | 58.2 | 20.6 | 22.5 |
| R-VGAE [IEEE TKDE'22] (Mrabah et al., 2022) | 71.3 | 49.8 | 48.0 | 44.9 | 19.9 | 12.5 | 69.2 | 30.3 | 30.9 |
| **Q-VGAE (Ours)** | 72.7 | 58.6 | 49.6 | 66.1 | 47.4 | 50.2 | 64.3 | 31.6 | 28.0 |
| VGAECD-OPT [Entropy'20] (Choong et al., 2020) | 27.2 | 37.3 | 22.0 | 51.8 | 25.1 | 15.5 | 32.2 | 25.0 | 26.1 |
| Mod-Aware VGAE [NN'22] (Salha-Galvan et al., 2022) | 67.1 | 52.4 | 44.8 | 51.8 | 25.1 | 15.5 | - | 30.0 | 29.1 |
| GMM-VGAE [AAAI'20] (Hui et al., 2020) | 71.9 | 53.3 | 48.2 | 67.5 | 40.7 | 42.4 | 71.1 | 29.9 | 33.0 |
| R-GMM-VGAE [IEEE TKDE'22] (Mrabah et al., 2022) | 76.7 | 57.3 | 57.9 | 68.9 | 42.0 | 43.9 | 74.0 | 33.4 | 37.9 |
| **Q-GMM-VGAE (Ours)** | 76.2 | 58.7 | 56.3 | 72.7 | 47.4 | 48.8 | 69.0 | 34.8 | 34.0 |

Table 1: Comparison of all methods on attributed datasets.

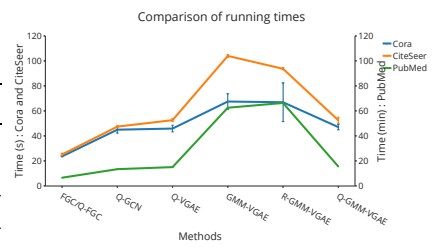

| Method | Brazil | | | Europe | | | USA | | |
|---|---|---|---|---|---|---|---|---|---|
| | ACC ↑ | NMI ↑ | ARI ↑ | ACC ↑ | NMI ↑ | ARI ↑ | ACC ↑ | NMI ↑ | ARI ↑ |
| GAE [NeurIPS'16] | 62.6 | 37.8 | 30.8 | 47.6 | 19.9 | 12.7 | 43.9 | 13.6 | 11.8 |
| DGI [ICLR'19] | 64.9 | 31.0 | 30.4 | 48.6 | 16.1 | 12.3 | 52.2 | 22.9 | 21.7 |
| GIC [PAKDD'21] | 40.5 | 23.5 | 14.1 | 40.4 | 9.4 | 6.2 | 49.7 | 22.1 | 19.9 |
| DAEGC [AAAI'19] | 71.0 | 47.4 | 41.2 | 53.6 | 30.9 | 23.3 | 46.4 | 27.2 | 18.4 |
| **Q-GCN (Ours)** | 51.1 | 31.9 | 23.7 | 45.5 | 30.8 | 25.1 | 43.8 | 19.1 | 14.8 |
| VGAE [NeurIPS'16] | 64.1 | 38.0 | 30.7 | 49.9 | 23.5 | 16.7 | 45.8 | 23.6 | 15.7 |
| **Q-VGAE (Ours)** | 50.1 | 35.0 | 19.8 | 46.6 | 19.5 | 17.5 | 46.2 | 19.5 | 16.9 |
| GMM-VGAE [AAAI'20] | 70.2 | 46.0 | 41.9 | 53.1 | 31.1 | 24.4 | 48.1 | 21.9 | 13.2 |
| R-GMM-VGAE [IEEE TKDE'22] | 73.3 | 45.6 | 42.5 | 57.4 | 31.4 | 25.8 | 50.8 | 23.1 | 15.3 |
| **Q-GMM-VGAE (Ours)** | 68.4 | 46.0 | 42.4 | 47.9 | 32.2 | 23.5 | 46.6 | 22.5 | 13.1 |

(a) Comparison of all methods on non-attributed datasets using degree.

(b) Comparison of running times of methods. Note that the scale for PubMed is in minutes (right axis), whereas for Cora and CiteSeer is in seconds.

## 5.5 ABLATION STUDIES

**Visualization of the latent space** First, we visualize how the latent space of the Q-VGAE and Q-GMM-VGAE changes over time for the Cora dataset. Plots for the rest of the datasets can be found in the supplementary material G. We use UMAP (Uniform Manifold Approximation and Projection) (McInnes et al., 2018) for dimensionality reduction from the latent space (64) to two dimensions.

**Comparison of running times** In Fig 2b we compare the running times of our method with other baselines. Our method take less than half as much time over all datasets. Especially on the largest dataset we have tested, PubMed, state-of-the-art methods GMM-VGAE and R-GMM-VGAE (un-modified) take about 60 minutes to complete whereas our Q-GMM-VGAE runs in under 15 minutes, a 75% reduction in running time, while also performing better. We also want to highlight that Q-FGC runs even faster, in just 6 minutes (for PubMed) and achieves 90% of the performance.

**Modularity Metric Comparison** We perform an experiment to see how much we gain in modularity over other baselines on Cora, CiteSeer and PubMed datasets. We report two types graph-based metrics, modularity $\mathcal{Q}$ and conductance $\mathcal{C}$, which don't require labels. Conductance measures the fraction of total edge volume that points outside the cluster. $\mathcal{C}$ is the average conductance across all clusters and a lower value is preferred. From Table 4a, we can see that even though DMoN (Tsitsulin et al., 2023) has the highest modularity, we achieve much higher NMI. Similarly for CiteSeer, we see a 40% improvement in NMI with only a 8% drop in the modularity, placing us closer to the ground truth. Moreover, we can also see that our methods perform better than the ones they were based on, i.e. Q-FGC > FGC, Q-VGAE > VGAE. Even though modularity is a good metric to optimize for, maximum modularity labelling of a graph does not always correspond to the ground truth labelling.

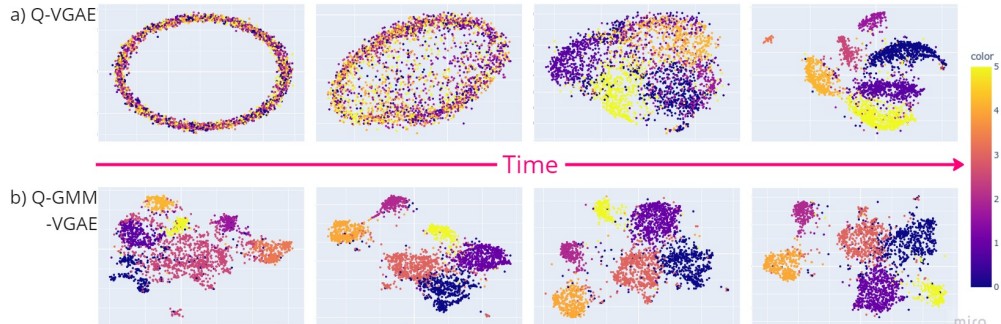

Figure 3: Evolution of the latent space of a) Q-VGAE and b) Q-GMM-VGAE over time for Cora. Colors represent cluster assignments.

| | Cora | | | CiteSeer | | | PubMed | | |
|---|---|---|---|---|---|---|---|---|---|
| | $\mathcal{C}\downarrow$ | $\mathcal{Q}\uparrow$ | NMI↑ | $\mathcal{C}\downarrow$ | $\mathcal{Q}\uparrow$ | NMI↑ | $\mathcal{C}\downarrow$ | $\mathcal{Q}\uparrow$ | NMI↑ |
| DMoN | 12.2 | **76.5** | 48.8 | 5.1 | **79.3** | 33.7 | 17.7 | **65.4** | 29.8 |
| FGC | 58.4 | 25 | 23.1 | 41.6 | 41.1 | 31 | 21.6 | 44.1 | 20.5 |
| Q-FGC | 13.3 | 72.5 | 51.7 | 16.8 | 64.9 | 40.16 | 26 | 40.3 | 28.1 |
| Q-GCN | 13.6 | 73.3 | 58.3 | 5.8 | 74.5 | 46.7 | **8.27** | 55 | 31.5 |
| VGAE | 17.6 | 60.8 | 38.1 | 12.8 | 55.8 | 21 | 13.5 | 45.8 | 26.9 |
| Q-VGAE | **9.5** | 71.5 | **58.4** | **4.6** | 72.4 | **47.3** | 9.4 | 52.12 | **31.8** |

(a) Comparison of modularity and conductance at the best NMI with DMoN. Note that DMoN is optimizing only modularity, whereas we are optimizing other important terms as well, as mentioned in Eqn 6, and thus gain a lot on NMI by giving up a small amount of modularity, making us closer to the ground truth.

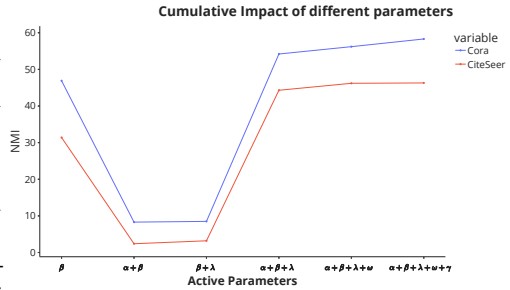

(b) Impact of active parameters on clustering performance. All terms in the loss equation 6 are represented by their parameters, for example the modularity term is represented by $\beta$. Additionally, $\omega$ represents the non-parameterized term.

For this reason, it is important to have the other terms in our formulation as well. By optimizing modularity, we can get close to the optimal model parameters (at which NMI would be 1), but will be slightly off-course. We can think of the other terms as correcting this trajectory.

**Stochastic Block Model (SBM) and variants** Please refer to supplementary material H.

**Importance of and Evolution of different loss terms** We analyze the evolution of the different loss terms during training, and also try to measure the impact of each term separately by removing terms from the loss one by one, as shown in Supplementary Material M. Also, we found that $||C\tilde{X} - X||_F^2$ is the most sensitive to change in its weight $\alpha$, followed by the terms related to $\gamma$, $\beta$ and then $\lambda$. This makes sense because if that constraint(relaxation) is not being met, then $C$ would have errors. Even though some of the terms do the heavy lifting, the other regularization terms do contribute to performance and more importantly, change the nature of $C$ : The term $\omega$ corresponds to smoothness of signals in the graph being transferred to the coarsened graph; this would affect $C$ by encouraging local "patches"/groups to belong to the same cluster. The term $\gamma$ ensures that the coarsened graph is connected - i.e. preserving inter-cluster relations, which simple contrastive methods destroy; this affects $C$ by making it so that $\Theta_C$ has minimal multiplicity of 0-eigenvalues.

## 6 CONCLUSION

In this paper, we have developed an optimization-based attributed graph clustering framework, Q-FGC, and its integration with deep learning-based architectures Q-GCN, Q-VGAE and Q-GMM-VGAE. We have performed graph clustering tasks using the proposed methods on real-world benchmark datasets and it is evident that incorporating modularity and graph regularizations into the coarsening framework improves the clustering performance. Also, integrating the proposed method with deep learning-based architecture improves the clustering performance by a significant amount. The proposed algorithms are provably convergent and much faster than state-of-the-art algorithms. A limitation of this method is that if for a graph, the ground truth labelling gives a low modularity, our method will be slower and not as stable convergence. However, it still manages to reach and surpass state-of-the-art methods on the Airports dataset, in which all graphs have a low modularity on the ground truth labels.

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
