$\uparrow$ | $\mathcal{C}\downarrow$ | $\mathcal{Q}\uparrow$ | NMI$\uparrow$ | $\mathcal{C}\downarrow$ | $\mathcal{Q}\uparrow$ | NMI$\uparrow$ |
| DMoN | 12.2 | **76.5** | 48.8 | 5.1 | **79.3** | 33.7 | 17.7 | **65.4** | 29.8 |
| FGC | 58.4 | 25 | 23.1 | 41.6 | 41.1 | 31 | 21.6 | 44.1 | 20.5 |
| Q-FGC | 13.3 | 72.5 | 51.7 | 16.8 | 64.9 | 40.16 | 26 | 40.3 | 28.1 |
| Q-GCN | 13.6 | 73.3 | 58.3 | 5.8 | 74.5 | 46.7 | **8.27** | 55 | 31.5 |
| VGAE | 17.6 | 60.8 | 38.1 | 12.8 | 55.8 | 21 | 13.5 | 45.8 | 26.9 |
| Q-VGAE | **9.5** | 71.5 | **58.4** | **4.6** | 72.4 | **47.3** | 9.4 | 52.12 | **31.8** |

(a) Comparison of modularity and conductance at the best NMI with DMoN. Note that DMoN is optimizing only modularity, whereas we are optimizing other important terms as well, as mentioned in Eqn 6, and thus gain a lot on NMI by giving up a small amount of modularity, making us closer to the ground truth.

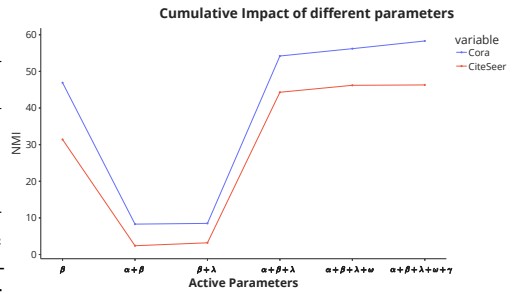

(b) Impact of active parameters on clustering performance. All terms in the loss equation 6 are represented by their parameters, for example the modularity term is represented by $\beta$. Additionally, $\omega$ represents the non-parameterized term.

For this reason, it is important to have the other terms in our formulation as well. By optimizing modularity, we can get close to the optimal model parameters (at which NMI would be 1), but will be slightly off-course. We can think of the other terms as correcting this trajectory.

**Stochastic Block Model (SBM) and variants** Please refer to supplementary material H.

**Importance of and Evolution of different loss terms** We analyze the evolution of the different loss terms during training, and also try to measure the impact of each term separately by removing terms from the loss one by one, as shown in Supplementary Material M. Also, we found that $||C\tilde{X} - X||_F^2$ is the most sensitive to change in its weight $\alpha$, followed by the terms related to $\gamma$, $\beta$ and then $\lambda$. This makes sense because if that constraint(relaxation) is not being met, then $C$ would have errors. Even though some of the terms do the heavy lifting, the other regularization terms do contribute to performance and more importantly, change the nature of $C$ : The term $\omega$ corresponds to smoothness of signals in the graph being transferred to the coarsened graph; this would affect $C$ by encouraging local "patches"/groups to belong to the same cluster. The term $\gamma$ ensures that the coarsened graph is connected - i.e. preserving inter-cluster relations, which simple contrastive methods destroy; this affects $C$ by making it so that $\Theta_C$ has minimal multiplicity of 0-eigenvalues.

## 6 CONCLUSION

In this paper, we have developed an optimization-based attributed graph clustering framework, Q-FGC, and its integration with deep learning-based architectures Q-GCN, Q-VGAE and Q-GMM-VGAE. We have performed graph clustering tasks using the proposed methods on real-world benchmark datasets and it is evident that incorporating modularity and graph regularizations into the coarsening framework improves the clustering performance. Also, integrating the proposed method with deep learning-based architecture improves the clustering performance by a significant amount. The proposed algorithms are provably convergent and much faster than state-of-the-art algorithms. A limitation of this method is that if for a graph, the ground truth labelling gives a low modularity, our method will be slower and not as stable convergence. However, it still manages to reach and surpass state-of-the-art methods on the Airports dataset, in which all graphs have a low modularity on the ground truth labels.

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

## A  PROOF OF LEMMA 2

When $\tilde{X}$ is kept constant, the optimization problem 6 gets reduced to:

$$\min_C f(C) = \text{tr}(\tilde{X}^T C^T \Theta C \tilde{X}) + \frac{\alpha}{2}\left\|C\tilde{X} - X\right\|_F^2 - \frac{\beta}{2e}\text{tr}(C^T BC) \tag{13}$$

$$- \gamma \log \det(C^T \Theta C + J) + \frac{\lambda}{2}\left\|C^T\right\|_{1,2}^2$$

subject to $C \in \mathcal{S}_c 1$ where, $J = \frac{1}{k}\mathbf{1}_{k \times k}$

$$\tag{14}$$

The term $\text{tr}(\tilde{X}^T C^T \Theta C \tilde{X})$ is convex function in $C$. This result can be derived easily using Cholesky Decomposition on the positive semi-definite matrix $\Theta$ (i.e. $\Theta = L^T L$):

$$\text{tr}(\tilde{X}^T C^T \Theta C \tilde{X}) = \text{tr}(\tilde{X}^T C^T L^T L C \tilde{X}) = \text{tr}((LC\tilde{X})^T LC\tilde{X}) = \left\|LC\tilde{X}\right\|_F^2 \tag{15}$$

Frobenius norm is a convex function, and the simplified expression is linear in C. Hence we can deduce that the tr(.) term is convex in C. The terms $\left\|C\tilde{X} - X\right\|_F^2$ and $\left\|C^T\right\|_{1,2}^2$ are convex because Frobenius norm and $l_{1,2}$ norm are convex in C.

Next, for the modularity term we show

$$\text{tr}(C^T BC) = \text{tr}(C^T B^{\frac{1}{2}} B^{\frac{1}{2}} C) = \text{tr}(C^T B^{T\frac{1}{2}} B^{\frac{1}{2}} C) = \left\|B^{\frac{1}{2}} C\right\|_F^2 \tag{16}$$

Hence, this is term is also convex in C.

For proving the convexity of $-\log\det(C^T \Theta C + J)$ we restrict function to a line. We define a function $g$:

$$g(t) = f(z + tu)\text{where, } t \in dom(g), z \in dom(f), u \in \mathbb{R}^n. \tag{17}$$

A function $f : \mathbb{R}^n \to \mathbb{R}$ is convex if $g : \mathbb{R} \to \mathbb{R}$ is convex.

The graph Laplacian matrix of the coarsened graph ($\Theta_c$) is symmetric and positive semi-definite having a rank of k-1. To convert $\Theta_c$ to positive definite matrix, we add a rank 1 matrix $J = \frac{1}{k}\mathbf{1}_{k \times k}$. $(\Theta_c + J = L^T L)$

$$f(L) = -\log\det(C^T \Theta C + J) = -\log\det(L^T L) \tag{18}$$

Now substituting L = Z + tU in the above equation.

$$g(t) = -\log\det((Z + tU)^T(Z + tU)) \tag{19}$$

$$= -\log\det(Z^T Z + t(Z^T U + U^T Z)t^2 U^T U) \tag{20}$$

$$= -\log\det(Z^T(I + t(UZ^{-1} + (UZ^{-1})^T) + t^2(Z^{-1})^T U^T UZ^{-1})Z) \tag{21}$$

$$\text{substituting } P = VZ^{-1} \tag{22}$$

$$= -(\log\det(Z^T Z) + \log\det(I + t(P + P^T) + t^2 P^T P)) \tag{23}$$

$$\text{Eigenvalue decomposition of} P = Q\Lambda Q^T \text{and} QQ^T = I \tag{24}$$

$$= -(\log\det(Z^T Z) + \log\det(QQ^T + 2tQ\Lambda Q^T + t^2 Q\Lambda^2 Q^T)) \tag{25}$$

$$= -(\log\det(Z^T Z) + \log\det(Q(I + 2t\Lambda + t^2\Lambda^2)Q^T)) \tag{26}$$

$$= -\log\det(Z^T Z) - \sum_{i=1}^{n} \log(1 + 2t\lambda_i + t^2\lambda^2) \tag{27}$$

Finding double derivative of $g(t)$:

$$g"(t) = \sum_{i=1}^{n} \frac{2\lambda_i^2(1+t\lambda_i)^2}{(1+2t\lambda_i+t^2\lambda_i^2)^2} \tag{28}$$

Since $g"(t) \geq 0 \forall\, t \in \mathbb{R}$, $g(t)$ is a convex function in $t$. This implies $f(L)$ is convex in $L$. We know that, $C^T\Theta C + J = L^T L$ so,

$$L = \Theta^{\frac{1}{2}}C + \frac{1}{\sqrt{kp}}\mathbf{1}_{p\times k} \tag{29}$$

Since $L$ is linear in $C$ and $f(L)$ is convex in $L$, $-\log\det(C^T\Theta C + J)$ is convex in C.

## B  OPTIMAL SOLUTION OF OPTIMIZATION OBJECTIVE

We first show that the function $f(C)$ is $L - Lipschitz$ continuous gradient function with $L = \max(L_1, L_2, L_3, L_4, L_5)$, where $L_1, L_2, L_3, L_4, and L_5$ are the Lipschitz constants of $\text{tr}(\tilde{X}^T C^T\Theta C\tilde{X}), \frac{\alpha}{2}\left\|C\tilde{X} - X\right\|_F^2, -\frac{\beta}{2e}\text{tr}(C^T BC), -\gamma\log\det(C^T\Theta C + J), and \frac{\lambda}{2}\left\|C^T\right\|_{1,2}^2$.

For the $\text{tr}(\tilde{X}^T C^T\Theta C\tilde{X})$ term, we apply triangle inequality and employ the property of the norm of the trace operator: $||tr|| = \sup\limits_{M\neq 0}\frac{|tr(M)|}{||M||_F}$.

$$|tr(\tilde{X}^T C_1^T\Theta C_1\tilde{X}) - tr(\tilde{X}^T C_2^T\Theta C_2\tilde{X})| \tag{30}$$

$$= |tr(\tilde{X}^T C_1^T\Theta C_1\tilde{X}) - tr(\tilde{X}^T C_2^T\Theta C_1\tilde{X}) + tr(\tilde{X}^T C_2^T\Theta C_1\tilde{X}) - tr(\tilde{X}^T C_2^T\Theta C_2\tilde{X})| \tag{31}$$

$$\leq |tr(\tilde{X}^T C_1^T\Theta C_1\tilde{X}) - tr(\tilde{X}^T C_2^T\Theta C_1\tilde{X})| + |tr(\tilde{X}^T C_2^T\Theta C_1\tilde{X}) - tr(\tilde{X}^T C_2^T\Theta C_2\tilde{X})| \tag{32}$$

$$\leq ||tr||\,||\tilde{X}^T(C_1-C_2)^T\Theta C_1\tilde{X}||_F + ||tr||\,||\tilde{X}^T C_2^T\Theta(C_1-C_2)\tilde{X}||_F \tag{33}$$

$$\leq ||tr||\,||\tilde{X}||_F||\Theta||\,||C_1-C_2||_F(||C_1||_F + ||C_2||_F) \text{ (Frobenius Norm Property)} \tag{34}$$

$$\leq 2\sqrt{p}||tr||\,||\tilde{X}||_F||\Theta||\,||C_1-C_2||_F \ \ (||C_1||_F = ||C_2||_F = \sqrt{p}) \tag{35}$$

$$\leq L_1||C_1-C_2||_F \tag{36}$$

The second term is $\frac{\alpha}{2}\left\|C\tilde{X} - X\right\|_F^2$ can be written as:

$$\frac{\alpha}{2}tr((C\tilde{X}-X)^T(C\tilde{X}-X)) \tag{37}$$

$$= \frac{\alpha}{2}tr(\tilde{X}^T C^T C\tilde{X} - X^T C\tilde{X} + X^T X - \tilde{X}^T C^T X) \tag{38}$$

$$= \frac{\alpha}{2}(tr(\tilde{X}^T C^T C\tilde{X}) - tr(X^T C\tilde{X}) + tr(X^T X) - tr(\tilde{X}^T C^T X)) \tag{39}$$

All the terms except $tr(X^T X)$ (constant with respect to C) in obtained in the expression will follow similar proofs to $\text{tr}(\tilde{X}^T C^T\Theta C\tilde{X})$.

Next we consider the modularity term:

$$|tr(C_1^T BC_1) - tr(C_2^T BC_2)| \tag{40}$$

$$= |tr(C_1^T BC_1) - tr(C_2^T BC_1) + tr(C_2^T BC_1) - tr(C_2^T BC_2)| \tag{41}$$

$$\leq |tr(C_1^T BC_1) - tr(C_2^T BC_1)| + |tr(C_2^T BC_1) - tr(C_2^T BC_2)| \tag{42}$$

$$\leq ||tr||\,||(C_1-C_2)^T BC_1||_F + ||tr||\,||(C_1-C_2)^T BC_2||_F \tag{43}$$

$$\leq ||tr||\,||B||\,||C_1-C_2||_F(||C_1||_F + ||C_2||_F) \text{ (Frobenius Norm Property)} \tag{44}$$

$$\leq L_3||C_1-C_2||_F \tag{45}$$

The Lipschitz constant for $-\gamma\log\det(C^T\Theta C + J)$ is linked to the smallest non-zero eigenvalue of the coarsened Laplacian matrix ($\Theta_c$) and is bounded by $\frac{\delta}{(k-1)^2}$ (Rajawat & Kumar, 2017), where $\delta$ is the minimum non-zero weight of $G_c$.

Lastly, in the term $\frac{\lambda}{2} \left\| C^T \right\|_{1,2}^2$ we can write $|C|_{ij} = C_{ij} \geq 0$ because $C \in \mathcal{S}_c$ and contains non-negative numbers.

$$\left\| C^T \right\|_{1,2}^2 = \sum_{i=1} p(\sum_{j=1} kC_{ij})^2 \tag{46}$$

$$= \sum_{i=1}^{p} ([C^T]_i \mathbf{1})^2 \tag{47}$$

$$= \|C\mathbf{1}\|_F^2 \qquad\qquad = tr(\mathbf{1}^T C^T C \mathbf{1}) \tag{48}$$

$tr(\mathbf{1}^T C^T C \mathbf{1})$ can be proved to be $L_5 - Lipschitz$ like the modularity and Dirichlet energy (smoothness) terms. This concludes the proof.

The majorized problem for L-Lipschitz and differentiable functions can now be applied. The Lagrangian of the majorized problem, 8 is:

$$\mathcal{L}(C, \tilde{X}, \mu) = \frac{1}{2}C^T C - C^T A - \mu_1^T C + \mu_2^T \left[ \left\| C_1^T \right\|_2^2 - 1, \cdots, \left\| C_i^T \right\|_2^2 - 1, \cdots, \left\| C_p^T \right\|_2^2 - 1 \right]^T \tag{49}$$

where $\mu = \mu_1 \| \mu_2$ are the dual variables and $A = \left( C - \frac{1}{L} \nabla f(C) \right)^+$

The corresponding KKT conditions (w.r.t $C$) are:

$$C - A - \mu_1 + 2[\mu_{2_o} C_0^T, \cdots, \mu_{2_i} C_i^T, \cdots, \mu_{2_p} C_p^T] = 0 \tag{50}$$

$$\mu_2^T \left[ \left\| C_1^T \right\|_2^2 - 1, \cdots, \left\| C_i^T \right\|_2^2 - 1, \cdots, \left\| C_p^T \right\|_2^2 - 1, \right]^T = 0 \tag{51}$$

$$\mu_1^T C = 0 \tag{52}$$

$$\mu_1 \geq 0 \tag{53}$$

$$\mu_2 \geq 0 C \qquad \geq 0 \tag{54}$$

$$\left\| [C^T]_i \right\|_2^2 \leq 1 \; \forall i \tag{55}$$

The optimal solution to these KKT conditions is:

$$C = \frac{(A)^+}{\sum_i \left\| [A^T]_i \right\|_2} \tag{56}$$

## C  PROOF OF CONVERGENCE

In this section, we prove that the sequence $\{\mathbf{C}^{t+1}, \tilde{\mathbf{X}}^{t+1}\}$ generated by Algorithm 1 converges to the set of Karush–Kuhn–Tucker (KKT) optimality points for Problem 6.

The Lagrangian of Problem 6 comes out to be:

$$\mathcal{L}(C, \tilde{X}, \mu) = \text{tr}(\tilde{X}^T C^T \Theta C \tilde{X}) + \frac{\alpha}{2} \left\| C\tilde{X} - X \right\|_F^2 - \frac{\beta}{2e} \text{tr}(C^T BC) \tag{57}$$

$$- \gamma \log \det(C^T \Theta C + J) + \frac{\lambda}{2} \left\| C^T \right\|_{1,2}^2 - \mu_1^T C + \sum_i \mu_{2i} \left[ \left\| C_i^T \right\|_2^2 - 1 \right] \tag{58}$$

where $\mu = \mu_1 \| \mu_2$ are the dual variables.

**w.r.t.** $C$, the KKT conditions are

$$2\Theta C\tilde{X}\tilde{X}^T + \alpha(C\tilde{X} - X)\tilde{X}^T - \frac{\beta}{e}BC - 2\gamma\Theta C(C^T\Theta C + J)^{-1} \tag{59}$$

$$+\lambda C\mathbf{1}_{k\times k} - \mu_1 + 2[\mu_{2_o}C_0^T, \cdots, \mu_{2_i}C_i^T, \cdots, \mu_{2_p}C_p^T] = 0$$

$$\mu_2^T\left[\left\|C_1^T\right\|_2^2 - 1, \cdots, \left\|C_i^T\right\|_2^2 - 1, \cdots, \left\|C_p^T\right\|_2^2 - 1\right]^T = 0 \tag{60}$$

$$\mu_1^T C = 0 \tag{61}$$

$$\mu_1 \geq 0 \tag{62}$$

$$\mu_2 \geq 0 \tag{63}$$

$$C \geq 0 \tag{64}$$

$$\left\|[C^T]_i\right\|_2^2 \leq 1 \; \forall i \tag{65}$$

Now, $C^\infty \equiv \lim_{t\to\infty} C^t$ is found from Equation 9 as:

$$C^\infty = C^\infty + \frac{1}{L}\left(2\Theta C^\infty\tilde{X}^\infty\tilde{X}^\infty + \alpha(C^\infty\tilde{X} - X)\tilde{X}^\infty - \frac{\beta}{e}BC^\infty \right. \tag{66}$$

$$\left. - 2\gamma\Theta C^\infty(C^{\infty T}\Theta C^\infty + J)^{-1} + \lambda C^\infty\mathbf{1}_{k\times k}\right)$$

$$0 = 2\Theta C^\infty\tilde{X}^\infty\tilde{X}^\infty + \alpha(C^\infty\tilde{X} - X)\tilde{X}^\infty - \frac{\beta}{e}BC^\infty \tag{67}$$

$$- 2\gamma\Theta C^\infty(C^{\infty T}\Theta C^\infty + J)^{-1} + \lambda C^\infty\mathbf{1}_{k\times k}$$

So, for $\mu = 0$, $C^\infty$ satisfies the KKT conditions.

**w.r.t.** $\tilde{X}$, the KKT conditions are:

$$2C^T\Theta C\tilde{X} + \alpha C^T(C\tilde{X} - X) = 0 \tag{68}$$

So, $X^\infty \equiv \lim_{t\to\infty} X^t$ found from Equation 11 will satisfy this as that equation is just a rearrangement of the KKT condition.

## D    DATASET SUMMARIES AND METRICS

Refer to 2 for the dataset summary.

**Metrics.** A pair of nodes are said to be in agreement if they belong to the same class and are assigned to the same cluster, or they belong to different classes and have been assigned different clusters. For a particular partitioning, ARI is the fraction of agreeable nodes in the graph. Accuracy is obtained by performing a maximum weight bipartite matching between clusters and labels. NMI measures the normalized similarity between the clusters and the labels, and is robust to class imbalances. Mutual Information between two labellings $X$ and $Y$ of the same data is defined as $MI(X, Y) = \sum_{i=1}^{|X|} \sum_{j=1}^{|Y|} \frac{|X_i \cap Y_i|}{N} \log \frac{N|X_i \cap Y_i|}{|X_i||Y_i|}$ and it is scaled between 0 to 1.

## E    TRAINING DETAILS

All experiments were run on an NVIDIA A100 GPU and Intel Xeon 2680 CPUs. We are usually running 4-16 experiments together to utilize resources (for example, in 40GB GPU memory, we can run 8 experiments on PubMed simultaneously). Again, the memory costs are more than dominated by the dataset. All experiments used the same environment running CentOS 7, Python 3.9.12, PyTorch 2.0, PyTorch Geometric 2.2.0.

## F VGAE

In a VGAE, the encoder learns mean ($\mu$) and variance ($\sigma$): $\mu = \text{GCN}_\mu(\mathbf{X}, \mathbf{A})$ and $\log \sigma = \text{GCN}_\sigma(\mathbf{X}, \mathbf{A})$ By using the reparameterization trick, we get the distribution of the latent space as: $q(\mathbf{Z}|\mathbf{X}, \mathbf{A}) = \prod_{i=1}^{N} q(\mathbf{z_i}|\mathbf{X}, \mathbf{A}) = \prod_{i=1}^{N} \mathcal{N}(\mathbf{z_i}|\mu_i, \text{diag}(\sigma_i^2))$ A common choice for decoder is inner-product of the latent space with itself which giving us the reconstructed $\hat{A}$. $p(\hat{\mathbf{A}}|\mathbf{Z}) = \prod_{i=1}^{p} \prod_{j=1}^{p} p(\hat{A}_{ij}|z_i, z_j)$, with $p(\hat{A}_{ij} = 1|z_i, z_j) = \text{sigmoid}(z_i^T z_j)$

## G PLOTS OF EVOLUTION OF LATENT SPACE FOR OTHER DATASETS

Refer to Figure 5. We can see the clusters forming in the latent space of the VGAEs. In the case of Q-VGAE, since a GCN is used on this space, it can learn non-linearities and the latent space shows different structures (like a starfish in CiteSeer). Moreover, these structures have their geometric centres at the origin and grow out from there. In contrast, for Q-GMM-VGAE, since a GMM is being learnt over the latent space, the samples are encouraged to be normally distributed in their independent clusters, all of which have different means and comparable standard deviations. So, we see multiple "blobs", which more or less follow a normal distribution. This plot effectively shows why a GMM-VGAE is more expressive than a VGAE.

| Name | p ($|\mathbf{V}|$) | n ($|\mathbf{x}_i|$) | e ($|E|$) | k ($y$) |
|---|---|---|---|---|
| Cora | 2708 | 1433 | 5278 | 7 |
| CiteSeer | 3327 | 3703 | 4614 | 6 |
| PubMed | 19717 | 500 | 44325 | 3 |
| Coauthor CS | 18333 | 6805 | 163788 | 15 |
| Coauthor Physics | 34493 | 8415 | 495924 | 5 |
| Amazon Photo | 7650 | 745 | 238162 | 8 |
| Amazon PC | 13752 | 767 | 491722 | 10 |
| ogbn-arxiv | 169343 | 128 | 1166243 | 40 |
| Brazil | 131 | 0 | 1074 | 4 |
| Europe | 399 | 0 | 5995 | 4 |
| USA | 1190 | 0 | 13599 | 4 |

Table 2: Datasets summary.

## H ATTRIBUTED SBM THEORY AND RESULTS

We validate the robustness and sensitivity of proposed methods to variance in the node features and graph structure. We are also generating features using a multivariate mixture generative model such that the node features of each block are sampled from normal distributions where the centers of clusters are vertices of a hypercube.

**SBM.** The Stochastic Block Model (SBM)(Nowicki & Snijders, 2001) is a generative model for graphs that incorporates probabilistic relationships between nodes based on their community assignments. In the basic SBM, a network with $p$ nodes is divided into $k$ communities or blocks denoted by $C_i$, where $i = 1, 2, \cdots, k$. The SBM defines a symmetric block probability matrix $B$ with size ($k \times k$), where each entry $B_{ij}$ represents the probability of an edge between a node in community $C_i$ and a node in community $C_j$. Diagonal entries of this matrix represents the probabilities of intra-cluster edges. This matrix $B$ captures the intra- and inter-community connections and is assumed to be constant. $P(i \leftrightarrow j|C_i = a, C_j = b) = B_{ab}$ denotes the probability of an edge existing between nodes $i$ and $j$ when node $i$ belongs to community $a$ and node $j$ belongs to community $b$. Using these probabilities, the SBM generates a network by independently sampling the presence or absence of an edge for each pair of nodes based on their community assignments and the block probability matrix $B$.

**Degree Corrected SBM.** DC-SBM(Karrer & Newman, 2011) takes an extra set of parameters $\theta_i$ controlling the expected degree of vertex $i$. Now, the probability of an edge between two nodes (using the same notation as above) becomes $\theta_i \theta_j B_{ab}$. This was introduced to handle the heterogeneity of real-world graphs.

**ADC-SBM Generation.** We make use of the `graph_tool` library to generate the DC-SBM adjacency matrix, with $p = 1000, k = 4$. To generate the $B$ matrix, we follow the procedure in (Tsitsulin et al., 2023), by taking expected degree for each node $d = 20$ and expected sub-degree

$d_{out} = 2$. This gives us $B$ as:

$$\begin{bmatrix} 18 & 2 & 2 & 2 \\ 2 & 18 & 2 & 2 \\ 2 & 2 & 18 & 2 \\ 2 & 2 & 2 & 18 \end{bmatrix}$$

Also, $\theta$ is generated by sampling a power-law distribution with exponent $\alpha = 2$. We constrain the generated vector to $d_{min} = 2$ and $d_{max} = 4$.

To generate features, we use the `make_classification` function in the `sklearn` library. We generate a 128-dimensional feature vector for each node, with no redundant channels. These belong to $k_f$ groups, where $k_f$ might not be equal to $k$. We test three scenarios: a) matched clusters ($k_f = k$) b) nested features ($k_f > k$) c) grouped features ($k_f < k$) as visualized in Figure 6. Note that for better visualization, `class_sep` was increased to 5 (however, the results are given with a value of 1, which is a harder problem).

**Results.** Our model is able to completely recover the ground truth labels (NMI/ARI/ACC = 1) under all the specified conditions.

## I    RESULTS ON VERY LARGE DATASETS

| Method | CoauthorCS | | | CoauthorPhysics | | | AmazonPhoto | | | AmazonPC | | | ogbn-arxiv | | |
|---|---|---|---|---|---|---|---|---|---|---|---|---|---|---|---|
| | ACC ↑ | NMI ↑ | ARI ↑ | ACC ↑ | NMI ↑ | ARI ↑ | ACC ↑ | NMI ↑ | ARI ↑ | ACC ↑ | NMI ↑ | ARI ↑ | ACC ↑ | NMI ↑ | ARI ↑ |
| FGC | 69.6 | 70.4 | 61.5 | 69.9 | 60.9 | 49.5 | 44.9 | 38.3 | 22.5 | 46.8 | 36.2 | 23.3 | 24.1 | 8.5 | 9.1 |
| **Q-FGC (Ours)** | 70.2 | 76.4 | 60.2 | 75.3 | 67.2 | 66.1 | 70.4 | 66.6 | **58.6** | **62.4** | 51 | 31.1 | 35.8 | 24.4 | 15.6 |
| **Q-GCN (Ours)** | 85.4 | 79.6 | 79.7 | 85.2 | **72** | **81.6** | 66.3 | 57.6 | 48.3 | 56.7 | 42.4 | 28.8 | 34.4 | 27.1 | 19.7 |
| **Q-VGAE (Ours)** | **85.6** | **79.9** | **81.6** | **86.7** | 69 | 77.7 | 69.0 | 59.4 | 49.0 | 62.3 | 45.7 | **47.2** | **39.5** | 30.4 | **24.7** |
| **Q-GMM-VGAE (Ours)** | 70.1 | 72.5 | 61.6 | 83.1 | 71.5 | 76.9 | **76.8** | **67.1** | 58.3 | 55.5 | **56.4** | 40 | OOM | OOM | OOM |
| DMoN | 68.8 | 69.1 | 57.5 | 45.4 | 56.7 | 50.3 | 61.0 | 63.3 | 55.4 | 45.4 | 49.3 | 47.0 | 25.0 | **35.6** | 12.7 |

Table 3: Comparison of methods on large attributed datasets.

## J    IMPLEMENTATION

The implementations for all the experiments can be found at `https://anonymous.4open.science/r/MAGC-8880/`.

We have extensively used the PyTorch(Paszke et al., 2019) and PyTorch Geometric(Fey & Lenssen, 2019) libraries in our implementations and would like to thank the authors and developers.

## K    EXPLANATION ON WHY VAE MANIFOLDS ARE CURVED

The latent space of a VAE is not constrained to be Euclidean. Connor et al. (2021) point out that the variational posterior is selected to be a multivariate Gaussian, and that the prior is modeled as a zero-mean isotropic normal distribution which encourages grouping of latent points around the origin. Works such as (Chen et al., 2020; Bogdanov & Shchur, 2021; Arvanitidis et al., 2018) make the VAE latent space to be Euclidean/Hyperbolic/Riemannian, and show good visuals. Moreover, it can be observed in our own work (Figure 3a and Appendix Figure 5a) that the latent manifolds are curved and so, are not suitable for conventional methods such as k-means clustering, which need Euclidean distance (3.2).

## L    COMPLEXITIES OF SOME GRAPH CLUSTERING METHODS

GCN*-based* clustering methods. Such as AGC(Zhang et al., 2019) - $\mathcal{O}(p^2nt + ent^2)$, R-VGAE(Mrabah et al., 2022) - $\mathcal{O}(pk^2n + (p(n + k) + e(p + k))$, S3GC(Devvrit et al., 2022) - $\mathcal{O}(pn^2s)$ [$s$ is average degree], HSAN(Liu et al., 2023b) - $\mathcal{O}(pBn)$ [they state it as $\mathcal{O}(B^2d)$ but it is only for 1 batch of size $B$ and not the whole epoch], VGAECD-OPT(Choong et al., 2020) - $\mathcal{O}(p^2nD^L)$ [where $D$ is the size of graph filter, $l$ is the number of linear layers] etc.

## M    EVOLUTION OF DIFFERENT LOSS TERMS THROUGHOUT TRAINING

Each separate series has been normalized by its absolute minimum value to see convergence behavior on the same graph easily. Every series is decreasing/converging (except gamma, which represents sparsity regularization and remains almost constant). Thus, we can be assured that no terms are counteracting and hurting the performance. The legend is provided in the graph itself. This plot is on the Cora dataset. 7

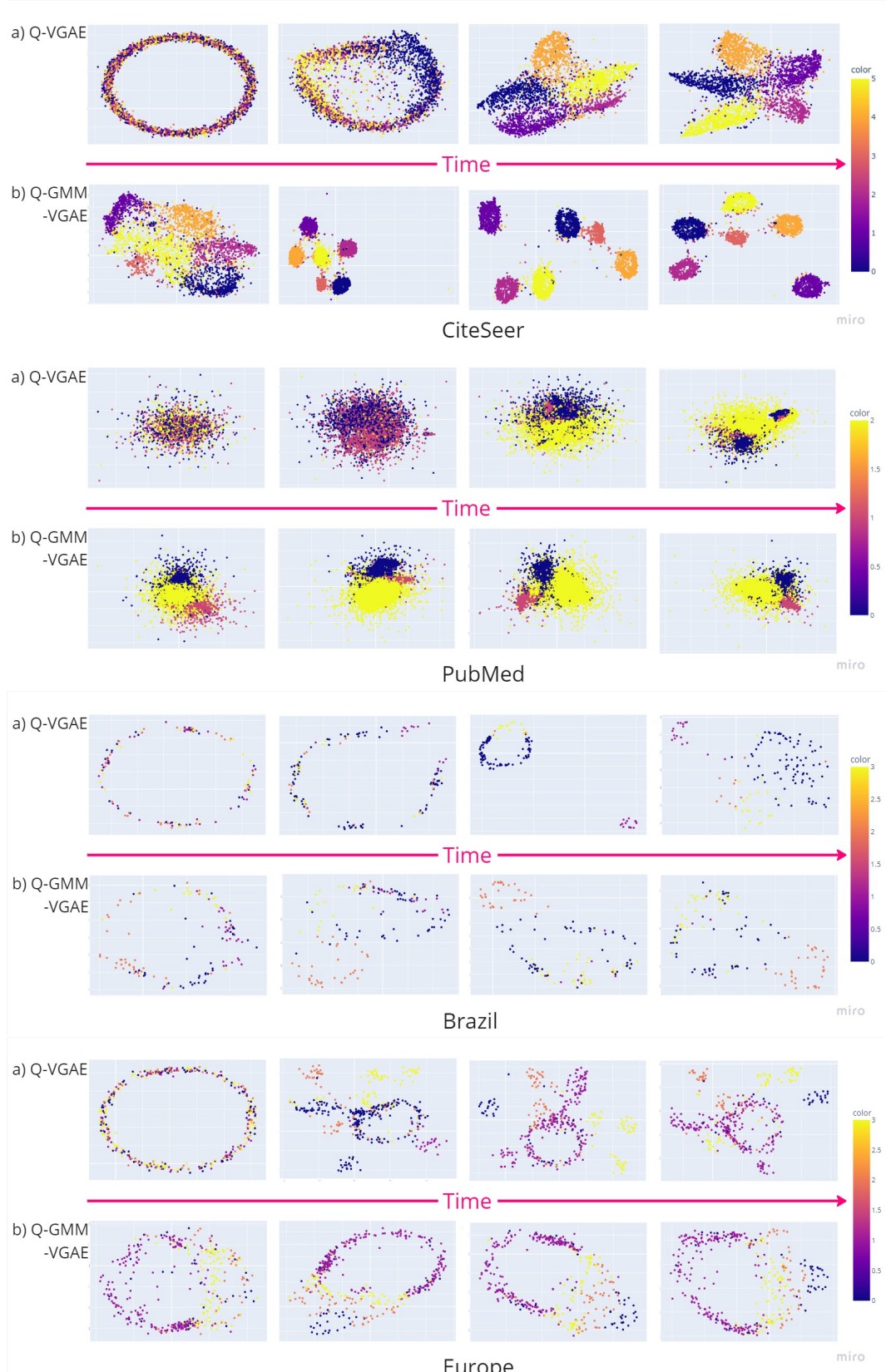

Figure 5: Plots of evolution of latent space for Q-VGAE and Q-GMM-VGAE methods for CiteSeer, PubMed, Brazil (Air Traffic) and Europe (Air Traffic) datasets.

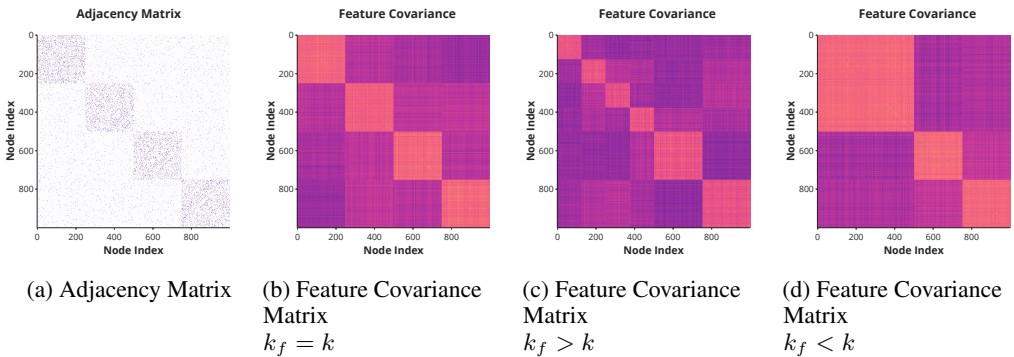

(a) Adjacency Matrix

(b) Feature Covariance Matrix $k_f = k$

(c) Feature Covariance Matrix $k_f > k$

(d) Feature Covariance Matrix $k_f < k$

Figure 6: Visualization of the generated adjacency and feature covariance matrices for the ADC-SBM

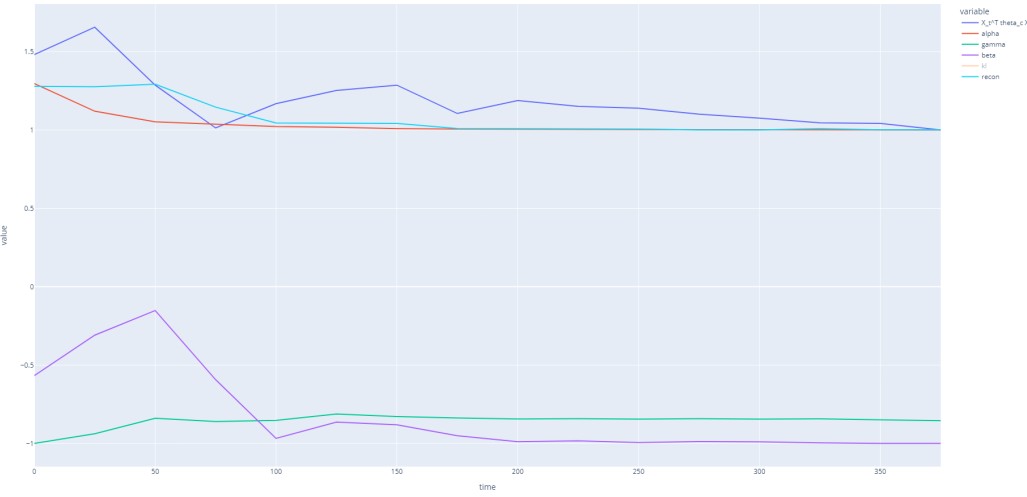

Figure 7