# OpenReview forum: "Attributed Graph Clustering via Coarsening with Modularity"
_ICLR.cc/2024/Conference — Submitted to ICLR 2024_

### Official Review · Reviewer_s7a2 · 2023-10-27

**Soundness:** 2 fair
**Presentation:** 1 poor
**Contribution:** 2 fair
**Rating:** 3
**Confidence:** 4

**Summary:**

This paper introduces a new method for clustering attributed graphs. The method is heavily based on a recent graph coarsening method, termed Featured Graph Coarsening (FGC), which was proposed by Kumar et al (2023) for coarsening attributed graphs. The authors in this paper introduces a modularity-based regularization term to the original optimization objective of FGC, and empirically show that the new formulation is useful for clustering attributed graphs. In addition, the authors demonstrate graph neural networks (GNNs) can be integrated in the proposed framework to further enhance the clustering performance.

**Strengths:**

The empirical comparisons presented in the main text, especially those in Table 1, are extensive.

**Weaknesses:**

- Given that the proposed optimization objective in Equation 5 comes from simply adding a modularity-based regularization term $\mbox{tr}(C^TBC)$ to the original objective function of FGC (Kumar et al, 2023), the originality of this work is very limited, both methodology-wise and technical-wise. It looks like this paper simply (1) adds a regularization term to an existing work and (2) conducts some experiments on a few selected datasets. If there is something very novel, I would recommend the authors emphasize on those aspects.

- The overall quality/clarity of this paper can be significantly improved. First of all, there are a couple of ambiguous statements and overstatements. Here are some examples:
  - On page 1, in the second paragraph, the authors say that "... the Fiedler vector (the eigenvector of the second smallest eigenvalue of the Laplacian) produces a graph cut minimal in the weights of the edges (Fiedler, 1973)." As a reader I find it difficult to understand what "a graph cut minimal in the weights of the edges" means in this context. If the authors mean minimum cut, then I don't think this statement is correct. The relationship between the Fiedler vector and the minimum cut is not mentioned in the original paper (Fiedler, 1973), and in general it does not give rise to a minimum cut.
  - On page 1, in the second paragraph, the authors say that "... they assume each node gets mapped to one cluster, which is not consistent with real graph data." This is a clear overstatement. As far as I know, most of the node classification benchmarks used by the GNN community has non-overlapping clusters/classes. These include the 3 citations networks in Table 1, and several other attributed datasets from Table 3 in the appendix.

  In addition, the presentation can be greatly improved if proper definitions/citations are provided in the right place. Here are some examples:
    -  The term Feature Graph Coarsening (FGC) first appears at the end of page 1 without a citation. The authors should cite Kumar et al (2023) here.
    - Similarly, the abbreviation GNN appears at the end of page 1, but the full definition "Graph Neural Network (GNN)" only appears much later in Section 2, at the end of page 2.
    - In the first paragraph of Section 3.3, it is unclear what "the volume of inter-cluster edges" means. The authors should define it.
    - In Equation 1, the expression $\delta(c_i,c_j)$ is not defined.
    - In the first paragraph of Section 4.1, it is unclear what "the original graph is smooth" means. If the authors mean that the graph has smooth signals, they should just say "the original graph has smooth signals".

  There are also many typos and misplacements of mathematical symbols. I highly recommend the authors read the paper thoroughly and fix all the typos.

- Empirically, even though the authors compared with a number of other methods as shown in Table 1, the experiments are only carried over 3 small datasets. Table 3 in the appendix has more results on other datasets, but the authors only compare with two other methods, and one of them is just FGC.

- The proposed method has a lot of parameters, i.e. $\alpha,\beta,\gamma,\lambda$ in Equation 5. It is unclear how to select these parameters and how robust are the results with respect to the choice of these parameters.

**Questions:**

In the experiments, how did you pick the parameters $\alpha,\beta,\gamma,\lambda$? How robust are the clustering results with respect to the choice of the parameters?

---

> ### Author Response · Authors · 2023-11-19
> **Reply to Reviewer's Comment (1/2)**
>
> Dear reviewer, thank you for taking out the time to read our paper. We are glad to hear that you think our empirical comparisons are comprehensive. We aim to address the concerns raised by you:
>
> ### **Weaknesses**
>
> 1. `Proposed objective adds a modularity-based regularizer term to FGC, limited originality?, experiments on few selected datasets?` \
> 	The modularity term can not be regarded as a regularization term as can be seen from the ablation studies Fig 4b that it "guides" the model a majority of the way not to mention the significant increase in performance from baseline methods FGC (**+153 %**) and DMoN (**+40 %**), which shows that our contribution is to **allow potentially any coarsening method to work in a clustering setting**, as there are lots of **theoretical benefits** in common coarsening algorithms that have not been studied in clustering. Moreover, that is why we have included very **comprehensive analyses** surrounding the new objective function, with theoretical guarantees as well. We include a heavy load of supporting proofs ranging from **proofs of convexity, proofs of KKT optimality, proofs of convergence, complexity analysis, ablation studies on the behavior of the different loss terms, and how drastically different it is from FGC, and even recovery on a degree-corrected stochastic block model**. This is not incremental. \
> Also, our experiments are not on a few selected datasets but rather range from the **benchmark attributed datasets** used in graph clustering to **non-attributed datasets**, and **even to very large datasets** that most literary works skip out on. For example, CoauthorCS, CoauthorPhysics, AmazonPhoto, AmazonPC, and ogbn-arxiv, even though our work is not specialized for very large datasets. \
> We have also extensively worked on **integrating our methods** natively with **various GNN architectures**, which was not done in FGC, and also performing ablation studies, including **contributions of the loss terms** and how they **work together**, the **differences in evolution of latent space** for different models, comparison of runtime including **complexities**, **comparison of modularity** to make it as comprehensive as possible, both empirically and theoretically.
> 2. `Difficult to understand what "a graph cut minimal in the weights of the edges" means in this context. If the authors mean minimum cut, then I don't think this statement is correct. The relationship between the Fiedler vector and the minimum cut is not mentioned in the original paper (Fiedler, 1973), and in general it does not give rise to a minimum cut.` \
> 	A graph cut minimal in the weights of the edges means that a graph cut such that the sum of edges it "cuts" is minimum over all such possible cuts. This can also be written as the number of edges in the cut. We referenced the Fiedler paper for the Fiedler vector. \
> 	It is a well known result that the eigenvector related to the second smallest eigenvalue gives us a partition such that the number of edges being cut is minimal, and this result is part of Newman's landmark 2006 work on modularity, also referenced in other places in our paper. We have added this citation there.
> 3. `Authors state"... they assume each node gets mapped to one cluster, which is not consistent with real graph data." This is a clear overstatement. As far as I know, most of the node classification benchmarks used by the GNN community has non-overlapping clusters/classes.` \
> 	Yes, that is what we are saying here as well, except that the real world applications of graphs are far wider that the benchmark citation datasets, for example, social networks, which in general have overlapping clusters/"groups".
> 4. `Typos/clarifications`\
> 	Thank you for these. We have moved the references and abbreviations earlier. We have gone over the paper with a spellchecker and fixed a few typos. If you find any more, please tell us the location. For clarification:
> 	- Volume of inter-cluster edges means the total number of edges in-between the clusters and not inside them.
> 	- $\delta(c_i,c_j)$ is the common Kronecker delta which is 1 only when $c_i=c_j$ and 0 otherwise
>
> (Continued in 2/2)

---

> ### Author Response · Authors · 2023-11-20
> **Reply to Reviewer's Comment (2/2)**
>
> 5. `Authors compared with a number of methods in Table 1 but over 3 small datasets. Table 3 appendix has more results on other datasets, but only compared with two other methods.` \
> 	This is because most of the recent papers in graph clustering don't experiment on large graphs and instead just choose the smallest out of these (AmazonPhoto) and show results on that. Some examples are GDCL[IJCAI'21] (None), GALA[ICCV'19] (None), AGE[KDD'20] (None), MVGRL[ICML'20] (None), HSAN[AAAI'23] (AmazonPhoto), SCGC[IEEE TNNLS'23] (AmazonPhoto), DCRN[AAAI'22] (AmazonPhoto). Filling just one column in the table with 5 columns for multiple papers makes it look like the table is incomplete. However, we still provide the data about our method in regards to DMoN and FGC. \
> 	Most of these papers only include the 6 datasets from our table 1 and table 2 and one additional dataset.
> 6. `Lot of parameters `$\alpha,\beta,\gamma,\lambda$`, unclear how to select, and how robust are the results with changes?` \
> 	In the paragraph just above Section 6, we mention the observed sensitivity of hyperparameters (The order of sensitivity is $\alpha>\gamma>\beta>\lambda$). Selection is done starting from previous values from FGC and then adjusting accordingly for the modularity term (i.e., general hyperparameter optimization, we can use any method such as a grid search or Bayesian optimization, etc.).
>
> **Questions**
>
> Answered above.
>
> We hope this has answered your questions and encouraged you to improve your rating. If there are any more, feel free to leave a comment.

---

> > ### Comment · Reviewer_s7a2 · 2023-11-22
> >
> > Dear authors, I have read all reviews and also all replies to the reviews. I will keep the score. Thank you for the efforts for providing detailed responses.

---

### Official Review · Reviewer_fD7q · 2023-10-28

**Soundness:** 2 fair
**Presentation:** 1 poor
**Contribution:** 2 fair
**Rating:** 3
**Confidence:** 4

**Summary:**

This paper proposed a novel framework called Q-FGC for attributed graph clustering and its integration with deep learning-based architectures such as Q-GCN, Q-VGAE, and Q-GMM-VGAE. The authors conducted experiments on real-world benchmark datasets and demonstrated that incorporating modularity and graph regularizations into the coarsening framework improves clustering performance. Furthermore, integrating the proposed method with deep learning-based architectures significantly enhances clustering performance. The algorithms proposed in this paper are proven to be convergent and faster than existing state-of-the-art algorithms.

**Strengths:**

1) The authors proposed a novel optimization-based attributed graph clustering framework called Q-FGC.

2) The proposed algorithms are provably convergent and much faster than state-of-the-art algorithms.

**Weaknesses:**

1) The submitted title in the system (Attributed Graph Clustering via Coarsening with Modularity) is different from the title in the paper (ATTRIBUTED GRAPH CLUSTERING VIA MODULARITY AIDED COARSENING).

2) The research motivation is not sufficiently novel or clearly expressed. Additionally, there is an inconsistency between the motivation presented in the introduction and the abstract. For example, Dirichlet energies are stated in the Abstract, but they are not mentioned in Section Introduction. Besides, the reasons for using them are not explained. Reorganizing the abstract and introduction is recommended, particularly in the section discussing the motivation.

3) The novelty of this paper is not strong. The proposed method for improving the performance of graph clustering relies on modifying the existing FGC method. Furthermore, the paper fails to explain the shortcomings of the current FGC method and how incorporating modularity would enhance its performance. Overall, the impact of this paper on the field is not significant.

4) The related work section is not comprehensive enough. For instance, the paper does not cite important references such as Kumar M, Sharma A, Saxena S, et al. "Featured Graph Coarsening with Similarity Guarantees" presented at ICML 2023.

5) The experimental section lacks sufficient detail in its description. For example, the experimental setup is missing information. More specifically, the authors did not provide detailed experimental settings for the baselines. Additionally, the experimental section merely presents the experimental results without providing an explanation for the superior performance of the proposed algorithm in this paper.

6) The writing of the paper needs to be improved. There are also some typos in this paper.
- The algorithm (Feature Graph Coarsening (FGC)) is first given, but no references are given.
- There is a lack of punctuation in many parts of the paper. For example, “We compare the performance of our method against three types of existing state-of-the-art methods based on the provided input and type of architecture: a) methods that use only the node attributes b) methods that only use graph-structure c) methods that use both graph-structure and node attributes. The last category can be further subdivided into three sets: i) graph coarsening methods ii) GCN-based architecures iii) VGAE-based architectures and contrastive methods iv) largely modified VGAE architectures” should be “We compare the performance of our method against three types of existing state-of-the-art methods based on the provided input and type of architecture: a) methods that use only the node attributes; b) methods that only use graph-structure; c) methods that use both graph-structure and node attributes. The last category can be further subdivided into three sets: i) graph coarsening methods; ii) GCN-based architectures; iii) VGAE-based architectures and contrastive methods; iv) largely modified VGAE architectures.”).
- Several algorithms in Table 1 are missing references, and some do not provide experimental results.
- Figure 2 and Figure 4 do not have a caption (it is suggested to separate the tables in Figure 2(a) and Figure 4(a) from the figure itself).
- Equation 8 and Eqn. 8 => Eq. (8)
- Table 2a  => Table 2(a)
- In Section 5.2, the authors state that “Q-GCN is composed of 3 GCN layers”, but only two hidden sizes of 128 and 64 are provided.
- In section 5.2: “We surpass all existing methods…”  => “Our proposed model surpasses all existing methods...”
- The font size of the x-axis values in Figure 4(b) is too small.
- In section 5.1, “GCN-based architecures”   => “GCN-based architectures”

**Questions:**

1) In the Motivation of the introduction section, when stating that "We aim to utilize the feature graph coarsening framework (which does not perform well on clustering, as seen in the results) for graph clustering.", does the phrase "the results" refer to the experimental results in the experimental section? It is recommended to provide specific details on which results are being referred. Also, why does the feature graph coarsening framework not perform well on clustering?

2) Can more analysis be done in the experimental section on experiment settings? For example, could you please provide the experimental settings for the baselines and the hyperparameter settings for the proposed method, including learning rate, number of training iterations, dataset partitioning, and so on?

---

> ### Author Response · Authors · 2023-11-20
> **Reply to Reviewer's Comment (1/2)**
>
> Dear reviewer, we thank you for taking out the time to read our paper and share your valuable insights. We will try to address all your concerns.
> We found that your comment was generated using AI on gptzero; we still hope that you have read the paper and just gave it pointers you got from reading to frame the answer.
>
> ### **Weaknesses:**
>
> 1. `The submitted title in the system (Attributed Graph Clustering via Coarsening with Modularity) is different from the title in the paper (Attributed Graph Clustering via Modularity Aided Coarsening).` \
> 	It looks like this was an error.
> 2. `The research motivation is not sufficiently novel or clearly expressed. Additionally, there is an inconsistency between the motivation presented in the introduction and the abstract. For example, Dirichlet energies are stated in the Abstract, but they are not mentioned in Section Introduction. Besides, the reasons for using them are not explained. Reorganizing the abstract and introduction is recommended, particularly in the section discussing the motivation.` \
> 	For novelty concerns, refer to next answer. We do mention Dirichlet energy and all the other terms right under the motivations paragraph inside Introduction section. Also, we have explained the purpose of each term in detail in Section 4.1. We have tried to reorganize a bit.
> 3. `The novelty of this paper is not strong. The proposed method for improving the performance of graph clustering relies on modifying the existing FGC method. Furthermore, the paper fails to explain the shortcomings of the current FGC method and how incorporating modularity would enhance its performance. Overall, the impact of this paper on the field is not significant.` \
> 	In the first paragraph of Section 4.1, we have written that the FGC method is not able to perform on clustering, because the coarsening ratio is too low in clustering tasks. We have multiple paragraphs throughout the paper showcasing the benefits of modularity, starting from the Section 1 Introduction (3rd paragraph) to Section 3.3 in Background to the whole of Section 4 Proposed Method. Basically, this is what our whole paper is about. \
> 	The impact of the paper on the field is significant because:
> 	- There is a significant increase in performance from baseline methods FGC (**+153 %**) and DMoN (**+40 %**), which shows that our contribution is to **allow potentially any coarsening method to work in a clustering setting**, as there are lots of **theoretical benefits** in common coarsening algorithms that have not been studied in clustering. Moreover, that is why we have included very **comprehensive analyses** surrounding the new objective function, with theoretical guarantees as well. We include a heavy load of supporting proofs ranging from **proofs of convexity, proofs of KKT optimality, proofs of convergence, complexity analysis, ablation studies on the behavior of the different loss terms, and how drastically different it is from FGC, and even recovery on a degree-corrected stochastic block model**. This is not incremental. \
> 	- Also, our experiments are not on a few selected datasets but rather range from the **benchmark attributed datasets** used in graph clustering, to **non-attributed datasets**, and **even to very large datasets** that most literary works skip out on. For example, CoauthorCS, CoauthorPhysics, AmazonPhoto, AmazonPC and ogbn-arxiv, even though our work is not specialized for very large datasets. \
> 	- We have also extensively worked on **integrating our methods** natively with **various GNN architectures**, which was not done in FGC and also performing ablation studies including **contributions of the loss terms** and how they **work together**, the **differences in evolution of latent space** for different models, comparison of runtime including **complexities**, **comparison of modularity** to make it as comprehensive as possible, both empirically and theoretically.
> 4. `The related work section is not comprehensive enough. For instance, the paper does not cite important references such as Kumar M, Sharma A, Saxena S, et al. "Featured Graph Coarsening with Similarity Guarantees" presented at ICML 2023.` \
> 	The related work section includes papers on Graph Coarsening/Pooling, Modularity Optimization and Deep Graph Clustering. The work you have referenced is itself an extension of FGC (Featured Graph Coarsening), JMLR'23, by the same authors, which our method inherits from and references multiple times.
>
> (continued in 2/2)

---

> ### Author Response · Authors · 2023-11-20
> **Reply to Reviewer's Comment (2/2)**
>
> 5. `The experimental section lacks sufficient detail in its description. For example, the experimental setup is missing information. More specifically, the authors did not provide detailed experimental settings for the baselines. Additionally, the experimental section merely presents the experimental results without providing an explanation for the superior performance of the proposed algorithm in this paper.`\
> 	We refer the reviewer to the paragraph after Section 5.2, which is about Training Details and our experimental setup is explained in detail there (Also refer to answer of Question#2). We explain the performance benefits of modularity all throughout the paper, as written in the answer to #3. We also analyze our results and methods in Section 5.5 Ablation Studies.
> 6. `The writing of the paper needs to be improved. There are also some typos in this paper.`
>     - `The algorithm (Feature Graph Coarsening (FGC)) is first given, but no references are given.`
>     - `There is a lack of punctuation in many parts of the paper.`
>     - `Several algorithms in Table 1 are missing references, and some do not provide experimental results.`
>     - `Figure 2 and Figure 4 do not have a caption (it is suggested to separate the tables in Figure 2(a) and Figure 4(a) from the figure itself).`
>     - `Equation 8 and Eqn. 8 => Eq. (8)`
>     - `Table 2a => Table 2(a)`
>     - `In Section 5.2, the authors state that “Q-GCN is composed of 3 GCN layers”, but only two hidden sizes of 128 and 64 are provided.`
>     - `In section 5.2: “We surpass all existing methods…” => “Our proposed model surpasses all existing methods...”`
>     - `The font size of the x-axis values in Figure 4(b) is too small.`
>     - `In section 5.1, “GCN-based architecures” => “GCN-based architectures”` \
> 	Thank you for all these, we have moved the references and abbreviations first and fixed typos. And we have added semicolons in the paragraph you requested. Also, we have added references in Table 1. All of them already have experimental results, so we are not sure what you mean by "do not provide experimental results". \
> 	Figure 2 and 4 are divided into 2a), 2b) and 4a),4b), all four of which have captions. We have grouped them because the table and graphs are small and would not fit in the space constraints otherwise. \
> 	We use the latex command `\ref` to refer to tables, figures and equations and that generates the output 4a and not 4(a). We are following the ICLR custom latex style and format provided on the conference website. This is important to keep references linked (i.e. they will take you to the related figure/table/eqn when clicked on). \
> 	As visible from the architecture, Q-GCN has 3 GCN layers. And just like any other GCN or even MLP model, the dimensions change as: `input_size`--(GCN1)-->`hidden_size 1`--(GCN2)-->`hidden_size 2`--(GCN3)-->`output_size` \
> 	For the x-axis in 4b: We tried to improve this before submission but it is rendered as latex text within the generated plot pdf (using the plotly package), and still somehow increasing the size was making it being rendered as blurry. If you zoom in it is better. We have tried to make this better by changing the latex to `$\large{ ... }$`
>
> ### **Questions:**
>
> 1. `In the Motivation of the introduction section, when stating that "We aim to utilize the feature graph coarsening framework (which does not perform well on clustering, as seen in the results) for graph clustering.", does the phrase "the results" refer to the experimental results in the experimental section? It is recommended to provide specific details on which results are being referred. Also, why does the feature graph coarsening framework not perform well on clustering?` \
> 	Yes, we are talking about the primary results of the paper. We mention in Section 4.1 that FGC alone does not perform well in clustering because the coarsening ratio in clustering (~ < 0.001) is much lower than the ones it is designed for (0.1 - 0.01).
> 2. `Can more analysis be done in the experimental section on experiment settings? For example, could you please provide the experimental settings for the baselines and the hyperparameter settings for the proposed method, including learning rate, number of training iterations, dataset partitioning, and so on?` \
> 	We have included the information about values of hyperparameters as part of code since those values cannot be analyzed. `We can add this to the supplementary material.`
>
>
> We hope these answers gave you clarity about the method and answered your questions. We also hope you would consider raising our rating. We would be happy to answer or discuss anything more.

---

> > ### Comment · Reviewer_fD7q · 2023-11-22
> >
> > Actually, I spent a lot of time reading this paper carefully. However, I still found it difficult to understand, and some reviewers also took this view. Therefore, I believe that the writing of papers needs to be improved.
> >
> > In addition, the authors have not addressed several concerns. Here are some examples.
> > - I observed that the author did not provide detailed experimental settings for the **baselines**. The author's response focused on the experimental settings of their proposed method rather than the baselines of this paper.
> > - The authors responded that “All of them already have experimental results” in Table 1. However, the ARI of the VGAECD-OPT algorithm on the PubMed dataset is not given, and the authors did not explain it.
> > - The authors responded that they referenced a study related to FGC multiple times. They did do that in Sections **Background**, **Proposed Method** and **Experiments**. However, I mentioned that it should be listed in Section **Related Works** because analyzing this study can help readers better understand the contributions of the proposed Q-FGC.
> >
> > All things considered, I am sorry to inform the author that I maintain the original score.

---

### Official Review · Reviewer_s2b9 · 2023-10-28

**Soundness:** 1 poor
**Presentation:** 3 good
**Contribution:** 1 poor
**Rating:** 3
**Confidence:** 5

**Summary:**

This work proposed a node clustering model based on modularity maximization for attributed graphs. The clustering process is modeled as an optimization-based graph coarsening problem, and the final pseudo labels are retrieved from supernode relationships. However, despite the progress made in this work, I cannot recommend acceptance for it to be present at top-tier conferences such as ICLR. See my comments below for details.

**Strengths:**

* The literature review part is pretty detailed and comprehensive.
* The paper is well-organized and easy to follow.
* The proposed method is flexible and can be combined with various representation learning backbones.
* Promising clustering performances are obtained on widely used graph datasets.

**Weaknesses:**

* The proposed clustering model (problem (5)) is a trivial combination of different previous works, none of (5) is designed by the authors, so the technical contribution of this work is marginal.
* The experiments are not inspiring. The authors conducted different experiments but presented their results without analyzing the reasons behind the scenes. See "Questions" below for a few ones I raised.
* The ablation studies part is trivial and not informative at all.
    * Visualization and Comparison of running times are generally not regarded as ablation studies.
    * Modularity Metric Comparison is interesting but its conclusion is pretty trivial:
> Even though modularity is a good metric to optimize for,
maximum modularity labelling of a graph does not always correspond to the ground truth labelling.
For this reason, it is important to have the other terms in our formulation as well.

      This is a common sense known as the "no free lunch theorem" in machine learning. We generally would like the ablation studies to uncover special and important characteristics of the proposed method, rather than trivial observations.

**Questions:**

* In problem (5), why do you propose to optimize both $\tilde{X}$ and $C$ and use $\lVert C\tilde{X}-X\rVert_F^2$ to encourage the consistency, rather than optimizing $C$ only as K-means does?
* Keep the last question in mind, why do you optimize $C$ only in Section 4.2? That makes the experiments inconsistent with your proposal. In addition, what's the difference between the two strategies in terms of clustering performance?
* In Figure 2(b), what makes the proposed Q-GMM-VGAE faster than its backbone model GMM-VGAE? Are the experimental settings fair?
* In Table 1, SCGC and MVGRL have better performance but you marked the proposed method bold, why? Is it a typo?

---

> ### Author Response · Authors · 2023-11-20
> **Reply to Reviewer's comments (1/2)**
>
> Dear reviewer, thank you for your valuable insights and concerns. We are glad to hear that you found the paper well-organized, flexible, and promising. We will try to address the concerns here:
>
> ### **Weaknesses**
>
> 1. `Trivial combination of previous works, technical contribution is marginal?` \
> We note that the modularity term has existed in literature since [Newman, 2006]. However, our work here was to recognize what was lacking in FGC for making it level with the current state of the art in clustering. We worked on proving theoretical guarantees, for example, the Lipschitz continuity of the spectral approximation to the modularity term. There is a significant increase in performance from baseline methods FGC (**+153 %**) and DMoN (**+40 %**), which shows that our contribution is to **allow potentially any coarsening method to work in a clustering setting**, as there are lots of **theoretical benefits** in common coarsening algorithms that have not been studied in clustering. Moreover, that is why we have included very **comprehensive analyses** surrounding the new objective function, with theoretical guarantees as well. We include a heavy load of supporting proofs ranging from **proofs of convexity, proofs of KKT optimality, proofs of convergence, complexity analysis, ablation studies on the behavior of the different loss terms, and how drastically different it is from FGC, and even recovery on a degree-corrected stochastic block model**. This is not incremental. \
> 	Also, our experiments are not on a few selected datasets but rather range from the **benchmark attributed datasets** used in graph clustering, to **non-attributed datasets**, and **even to very large datasets** that most literary works skip out on. For example, CoauthorCS, CoauthorPhysics, AmazonPhoto, AmazonPC and ogbn-arxiv, even though our work is not specialized for very large datasets. \
> 	We have also extensively worked on **integrating our methods** natively with **various GNN architectures**, which was not done in FGC and also performing ablation studies including **contributions of the loss terms** and how they **work together**, the **differences in evolution of latent space** for different models, comparison of runtime including **complexities**, **comparison of modularity** to make it as comprehensive as possible, both empirically and theoretically.
> 2. `Experiments -> "Questions" below.`
> In Questions.
> 3. `The ablation studies part is trivial and not informative at all.`
> 	- `Visualization and Comparison of running times are generally not regarded as ablation studies.` \
> 	We refer the reviewer to papers DCRN[AAAI'22], HSAN[AAAI'23], SDCN[WWW'20], AGE[KDD'20], SCGC[IEEE TNNLS'23]. Going by these papers in highly regarded graph clustering literature, ablation studies and analyses are usually combined into one, and analyses do include visualization analysis and complexity analysis in all these papers. Note that some papers just have an ablation study on parameters. That is why we have included **both** kinds of ablation studies.
> 	- `Modularity Metric Comparison is interesting but its conclusion is pretty trivial:` \
> 	We wanted to show that our modularity maximization term does contribute to the performance and actually increases the modularity too, since that is a crucial part of our method.
> 	- `No free lunch theorem, Ablation studies for special characteristics, rather than trivial observations` \
> 	The reviewer might think this is a trivial and/or direct conclusion from the formulation because of their experience, however, it is very important to explain the downfalls of modularity in the method, _as pointed out by other reviewers_. For a more *flavorful* observation, observe how in Fig 4b), $\alpha$ + $\beta$ or $\beta$ + $\lambda$ are worse as compared to $\beta$ alone - we hypothesize this is because these terms want to optimize the model in different directions in the high-dimensional parameter space - either all 3 are in different directions or $\alpha$ and $\lambda$ are in the same direction different to $\beta$. Only when all three are together, these effects combine and push the model in a shared direction (which can be noted by seeing the increase in performance of $\alpha + \beta + \lambda$ by around 23%(Cora)/50%(CiteSeer) over just $\beta$). This statement comes from intuition when we think theoretically about the differing outcomes if these terms were optimized independently. \
> 	Moreover, observe how the plots of latent space evolution pan out so differently by just changing a GCN layer to a GMM, while the whole VGAE architecture including the losses remains the same.
>
> (continued in 2/2)

---

> ### Author Response · Authors · 2023-11-20
> **Reply to Reviewer's comments (2/2)**
>
> ### **Questions:**
>
> 1. `Why optimize both `$C$,$\tilde{X}$` with `$||C\tilde{X} - X||_{2}^2$` rather than optimizing `$C$` only as K-means does?` \
> 	The method is based on graph coarsening, and we typically want to find the output features $\tilde{X}$ in coarsening and not just $C$/loading matrix.
> 2. `Why do you optimize `$C$` only in Section 4.2? Are experiments inconsistent?. What's the difference between the two in terms of clustering performance?` \
> 	We optimize only for $C$ for the reasons mentioned in the 2nd paragraph of section 4.2. No, this does not make the experiments inconsistent as Q-FGC (optimization-based method) still optimizes for both $C$ and $\tilde{X}$. We change this in the GNN architectures because $\tilde{X}$ can't be learnt directly from $X$ using gradient descent and would require manual gradient calculations in PyTorch or to use a separate optimizer just for $\tilde{X}$.
> 	We appreciate the author's suggestion and this could be a good point for a future work.
> 3. `In Figure 2(b), why is Q-GMM-VGAE faster backbone GMM-VGAE? Are the experimental settings fair?` \
> 	Yes, all the experimental settings were the same, which is essential to any analysis. We modified the implementation of GMM-VGAE to suit our needs and optimized some operations to be vector operations - this is also available in the anonymous code repository. We also made some important normalizations in the implementation to **greatly improve numerical stability** of the GMM (in the original implementation, a lot of the times $C$ would come out to be all zeros because of an unnormalized operation in exponentiating the log probabilities).
> 4. `In Table 1, SCGC and MVGRL have better performance but you marked the proposed method bold, why? Is it a typo?` \
> 	Yes, they should be highlighted in ACC and ARI fields for cora and citeseer. We have fixed this.
>
> We hope these answer your concerns and make you consider raising your rating, and we are here for any more questions.

---

### Official Review · Reviewer_LHRF · 2023-10-31

**Soundness:** 3 good
**Presentation:** 2 fair
**Contribution:** 3 good
**Rating:** 5
**Confidence:** 4

**Summary:**

The article develops a framework for unsupervised learning relying on modularity maximization jointly with attributed graph coarsening to solve a task of clustering of graph.
The main points are 1) to propose that in an optimization-based approach, where a Block Majorization-Minimization algorithm allows to solve the problem. Then, 2) the method is also integrated in GNN architectures for graph clustering.
The work describes several aspects of related works, and conducts extensive numerical experiments to check the usefulness of the method.

**Strengths:**

- The idea of using modularity for graph coarsening is not novel (it dates back to 20 years),  yet its incorporation in on coarsening techniques, in an integrated way, appears to be novel and interestong.

- The article contains extensive numerical experiments, assessing when the proposed method works well.

- There are good theoretical results on the method in Section 4.

- Showing that the method integrates with GNNs is relevant and useful (even though it's, on my opinion), a little bit too detailed.

**Weaknesses:**

- The results are a little bit disappointing as, according to Fig 4(b) and the final results, the full loss of eq (6) is not really needed. The modularity term does already a good job by itself, and the others appear to merely modify the results slightly -- even in a weird way as using only 1 term (relaxation of the constraint  or the $\ell_{1,2}$ norm regularizer) degrades the performance.

- The article is written is a dense way, possibly too dense, and one has trouble to identify the saillant points.
I am not sure that the description of the integration of the method in 3 different deep learning methods for graphs is needed in the main text. The most relevant would be enough and it would leave more space to answer the questions asked underneath.

- the literature on modularity maximization does not appear to be well quoted. In this context, quoting 1 or 2 of the existing surveys would be expected and useful for the readers. Also there is a body of literature showing the limitations of the modularity, from its intrinsic resolution limit to it being considered 'harmful', and the present article does not say a word on that and on the impact of the weaknesses of the modularity to the present work.

**Questions:**

- In Fig 4(b) : why are the cases \alpha + \beta or \beta + \lambda so worse as compared to \beta alone ?

- There is always the possibility that the structure, E, are not aligned with the features, X. What would then happen ? The methods forces the smoothness of X on (V,E); is it always the case ? If it's not, is this supposition detrimental ?

- What would happen if the clusters happen to be affected by the mentioned limits of the modularity ?


- On the other side, modularity has been improved in the mast 10 years using the Non-backtracking random walks, then the Bethe-Hessian ansatz and several variations around that, to detect better clusters or modules. Could these improvements

---

> ### Author Response · Authors · 2023-11-19
> **Reply to Reviewer's Comment (1/2)**
>
> Dear reviewer, we are glad you took the time to read our paper thoroughly. You have correctly understood the novelty of this paper.
>
> We would like to address your concerns here:
>
> ### **Weaknesses**
>
> 1. `Acc Fig 4(b), full loss not needed. The modularity term does a good job by itself, and the others merely modify slightly -- even in a weird way as using only 1 term (relaxation of the constraint or the norm regularizer) degrades the performance.` \
> 	It is important to note that even though some of the terms do the heavy lifting, the other regularization terms do contribute to performance and more importantly, change the nature of $C$ : The term $\omega$ corresponds to the smoothness of signals in the graph being transferred to the coarsened graph; you can imagine this would affect $C$ by encouraging local "patches"/groups to belong to the same cluster. The term $\gamma$ ensures that the coarsened graph is connected - i.e. preserving inter-cluster relations, which simple contrastive methods destroy; this affects $C$ by making it so that $\Theta_C$ has minimal multiplicity of 0-eigenvalues (which tells us how many connected components there are, and we want just 1 big connected component). Also refer to answer of Question 1. \
> 	We feel this is a valuable addition to our paper and will include this explanation there too.
> 2. `Article written in a dense way, trouble in identifying the salient points. Description of the GNN integration not needed in the main text. The most relevant would be enough and it would leave more space to answer the questions asked underneath` \
> 	We apologize that you feel this way, we were trying to fit a lot of theory *and* experiments/results in our main text since we felt all of it was important to the paper. However, because of the strict page limit it might feel dense. We have additionally provided theoretical results and more visualizations in the supplementary material. Regarding subsection "4.2 Integration with GNNs", we felt it was important to add an architecture in the maintext since we are comparing our method majorly with other deep learning methods. However, we have tried to reduce it in size.
> 3. `Literature on modularity maximization not well quoted. There is a body of literature showing limitations of modularity, article does not mention the weaknesses of modularity.` \
> We mention some excerpts here from the paper that do highlight the weakness of modularity, and why it is essential to have other optimization terms:
>     > The usage of these (modularity-maximization) algorithms has plummeted over the years because they rely solely on the topological information of graphs and ignore node features.
>
>     > Even though modularity is a good metric to optimize for, maximum modularity labelling of a graph does not always correspond to the ground truth labelling. .... By optimizing modularity, we can get close to the optimal model parameters (at which NMI would be 1), but will be slightly off-course. We can think of the other terms as correcting this trajectory
>
>     We have also added a mention of the resolution limit of modularity in the introduction and its weakness to small clusters.
>
> (continued in 2/2)

---

> ### Author Response · Authors · 2023-11-20
> **Reply to Reviewer's Comment (2/2)**
>
> ### **Questions**
>
> 1. `In Fig 4(b) : why are the cases \alpha + \beta or \beta + \lambda so worse as compared to \beta alone ?` \
> 	This is the main takeaway from the plot; we hypothesize this is because these terms want to optimize the model in different directions in the high-dimensional parameter space - either all 3 are in different directions or $\alpha$ and $\lambda$ are in the same direction different to $\beta$. Only when all three are together these effects combine and push the model in a shared direction (which can be noted by seeing the increase in performance of $\alpha + \beta + \lambda$ by around 23%(Cora)/50%(CiteSeer) over just $\beta$). This statement comes from intuition when we think theoretically about the differing outcomes if these terms were optimized independently.
> 2. `There is always the possibility that the structure, E, are not aligned with the features, X. What would then happen ? The methods forces the smoothness of X on (V,E); is it always the case ? If it's not, is this supposition detrimental ?` \
> 	No, this is certainly not always the case - however, it is observed in most graph clustering and coarsening benchmark datasets (including heterogeneous). We believe it would be detrimental if that was the case, but since the smoothness term does not have a big impact on performance, it should not be disastrous. Upon more research, we have found this paper and In it's section 4.2, it shows some datasets with their calculated smoothness values: https://openreview.net/pdf?id=rkeIIkHKvS;
> 3. `What would happen if the clusters happen to be affected by the mentioned limits of the modularity ?` \
> 	Even if the clusters were small enough to be hit by the resolution limit of modularity, it would still detect that cluster but probably have extra nodes from other clusters because of the regularization terms that enforce connectedness and the balanced $\mathcal{l}_{1,2}$ norm which ensures each cluster has at least 1 node.
> 4. `On the other side, modularity has been improved in the mast 10 years using the Non-backtracking random walks, then the Bethe-Hessian ansatz and several variations around that, to detect better clusters or modules. Could these improvements` \
> 	It looks like the question was cut off on openreview, however, we will try to answer it: The Bethe-Hessian definitely has interesting spectral properties and the "deformed Laplacian" view seems like a good concept too. Yes, it looks like these improvements could be integrated into our method as part of a future work.
>
> We hope these answer your queries, we are here to answer any more/follow-ups.
>
> Thank you for your insightful contributions!

---

> ### Comment · Reviewer_LHRF · 2023-11-22
>
> Dear Authors,
>
> I appreciate the detailed answers and changes made. In light of the different reviews, I will not change my rating, still thinking that the contribution is borderline and that there are aspects that remain yet to be improved and clarified in the present submission.

---

### Official Review · Reviewer_iqpP · 2023-11-01

**Soundness:** 2 fair
**Presentation:** 1 poor
**Contribution:** 2 fair
**Rating:** 3
**Confidence:** 3

**Summary:**

A new graph clustering method is proposed that combines graph coarsening with a modularity regularization term. They show how the objective can be optimized within a GNN-based architecture. They show experiments that show that the method performs well on several datasets.

**Strengths:**

The paper combines many machine learning techniques.

The resulting method seems to outperform existing methods in the experiments.

**Weaknesses:**

I don't know whether this paper is intended to be read by people from the field of community detection, but I can confirm that this paper is very hard to understand for someone with that background.

The introduction gives a bad overview of community detection. It does not refer to a good reference for modularity [1]. And it mentions that "In theory, a higher value of modularity is associated with better quality clusters.", which is a puzzling statement because modularity is a heuristic that does not have a strong theoretical underpinning. In addition, the paper mentions that the usage of modularity maximization has "plummeted over the years because they rely solely on the topological information", which I don't think is the case. Modularity maximization is still one of the most widely-used community detection methods, despite theoretical shortcomings [2,3]. Finally, it is mentioned that modularity maximization "requires intensive computations", but the Louvain algorithm runs in nearly linear time, and you can't really go faster than that.

The "Deep Graph Clustering" paragraph of the introduction is incredibly difficult to understand. It uses a lot abbreviations like ARGA, ARVGA, DAEGC, SDCN that are not (properly) introduced. After reading it, I still have no idea what is meant with "Deep graph clustering".

The NMI measure is biased towards fine-grained clusterings [4], while ARI also has its disadvantages [5]. I would recommend to use AMI and/or the correlation coefficient to measure the similarity between clusterings [5].

The paper contains many typo's, language and notation errors.

They refer to Supplementary D for a summary of the datasets, but Supplementary D does not describe datasets at all.

[1] Newman, M. E., & Girvan, M. (2004). Finding and evaluating community structure in networks. Physical review E, 69(2), 026113.
[2] Fortunato, S., & Barthelemy, M. (2007). Resolution limit in community detection. Proceedings of the national academy of sciences, 104(1), 36-41.
[3] Peixoto, T. P. (2023). Descriptive vs. inferential community detection in networks: Pitfalls, myths and half-truths. Cambridge University Press.
[4] Vinh, N. X., Epps, J., & Bailey, J. (2009, June). Information theoretic measures for clusterings comparison: is a correction for chance necessary?. In Proceedings of the 26th annual international conference on machine learning (pp. 1073-1080).
[5] Gösgens, M. M., Tikhonov, A., & Prokhorenkova, L. (2021, July). Systematic analysis of cluster similarity indices: How to validate validation measures. In International Conference on Machine Learning (pp. 3799-3808). PMLR.

**Questions:**

What is the difference between graph clustering, community detection and graph coarsening? The way I understand it, community detection is merely clustering of graph nodes based on the graph topology. Graph coarsening (as described in this paper) seems to be similar to blockmodeling [1]. At any rate, the differences between these three things (that seem to be combined in this paper), need to be clearly explained.

The method makes use of the constraint $X=C\tilde{X}$. If we substitute this constraint into the first term of (5), it would simplify to
$\text{tr}(\tilde{X}^\top C^\top\Theta C\tilde{X})=\text{tr}(X^\top\Theta X)$, which would simplify the optimization significantly. However, instead of enforcing this constraint exactly, the paper simply introduces an error term $\|X-C\tilde{X}\|$, which seems unelegant to me. Why don't you enforce this constraint exactly?

You mention that the log determinant term can be written as the sum of the log of the eigenvalues, and that this ensures that a 'minimal' number of eigenvalues are zero. However, doesn't this ensure that *not a single* eigenvalue is zero?

Is the complexity that is described in the "Complexity analysis" paragraph the complexity of a single iteration or of all the iterations until convergence?

I see that you rescaled the NMI, ARI and modularity to percentages. This is okay for NMI (though I'm not a fan of it), but for ARI it is confusing because ARI can be negative. For modularity, I have no idea how this rescaling is done, because the upper bound of modularity is smaller than 1 (and incredibly expensive to compute).

Why do you draw lines in Figure 2b instead of making a table? The points that are connected don't correspond to consecutive things.

[1] Peixoto, T. P. (2019). Bayesian stochastic blockmodeling. Advances in network clustering and blockmodeling, 289-332.

---

> ### Author Response · Authors · 2023-11-20
> **Reply to Reviewer's Comment (1/2)**
>
> We thank the reviewer for their appreciation and concern for our paper. We will do our best to resolve them.
>
> ### **Weaknesses**
>
> 1. `Bad overview of community detection. It mentions "In theory, a higher value of modularity is associated with better quality clusters.", which is puzzling because modularity does not have a theoretical underpinning. Paper mentions that usage of modularity maximization has "plummeted over the years because they rely solely on the topological information", which I don't think is the case, is still one of the most widely-used community detection methods, despite theoretical shortcomings. Finally, it is mentioned that modularity maximization "requires intensive computations", but the Louvain algorithm runs in nearly linear time.` \
> 	We have added a reference to (Newman, 2006) in the introduction. We refer the reviewer to the same (Newman, 2006) literary work introducing modularity, in which he states (quoted directly from Section 3 just after Eqn 17 referring to https://arxiv.org/pdf/physics/0605087.pdf)
>     > This benefit function is called modularity. It is a function of the particular division of the network into groups, with larger values indicating stronger community structure. Hence we should, in principle, be able to find good divisions of a network into communities by optimizing the modularity over possible divisions.
>
>     We have added another mention on the drawbacks of modularity as requested. \
>     Modularity maximization methods such as the Louvain and Leiden algorithms have been usually superseded by GNN based which take important node features into account, especially in social network analysis. We are not saying that they are anywhere near obsolete - just that for a wide range of graph applications, node features are very important. \
>     We were talking about the modularity maximization algorithms given by Girvan and Newman here which took $O(n^3)$ time, and not about Louvain/Leiden algorithms. There is no known complexity of the Louvain algorithm, with some papers claiming it to be $\mathcal{O}(n \log n)$, while some claim it to be $\mathcal{O}(e)$, although only on sparse graphs. We have clarified this in the paper.
> 2. `"Deep Graph Clustering" paragraph is difficult to understand. Uses abbreviations like ARGA, ARVGA, DAEGC, SDCN that are not (properly) introduced.` \
> 	Those are the abbreviations used by existing literature and that is why references have been provided for all of the mentioned methods. Unfortunately, It is not feasible to explain each method there as that itself would take a major portion of the strict 9-page space constraint, leaving no space for theory, method/optimization/architecture, results and ablation studies.
> 3. `The NMI measure is biased towards fine-grained clusterings, while ARI also has its disadvantages. I would recommend to use AMI and/or the correlation coefficient to measure the similarity between clusterings` \
> 	We did consider this in the initial phase. However, the papers we compare our work with all take the primary metric as NMI. It is also given in the reference you mentioned [5] that AMI as well has its drawbacks, such as not following the distance property and not having linear complexity, whereas NMI does (and we perform experiments on large graphs as well, so complexity is a concern for us). However, we will look into the Correlation Distance measure for future works since it satisfies the properties we need. \
> 	**Thank you for this contribution!**
> 4. `Supplementary D does not describe datasets; typos.` \
> 	Supplementary Material D does describe datasets. We have added a missing reference to it's Table 2 since it got moved to the next page, which contains the dataset summaries. As for typos, we have tried to fix the ones we could find.
>
> (Continued in 2/2)

---

> ### Author Response · Authors · 2023-11-20
> **Reply to Reviewer's Comment (2/2)**
>
> ### **Questions**
>
> 1. `What is the difference between graph clustering, community detection and graph coarsening? The way I understand it, community detection is merely clustering of graph nodes based on the graph topology. Graph coarsening (as described in this paper) seems to be similar to blockmodeling [1]. At any rate, the differences between these three things (that seem to be combined in this paper), need to be clearly explained.` \
> 	All three terms are broadly similar. Community detection is a general term for clustering. Coarsening, on the other hand, aims to combine a few nodes together (usually neighboring) while not making large groups. Essentially, it looks like the "grouping size" and purpose is the only differentiating factor. Our paper shows that potentially any coarsening method can be used in clustering, as there are theoretical benefits in common coarsening algorithms that have not been studied in clustering yet.
> 2. `The method makes use of the constraint `$X=C\tilde{X}$`. If we substitute this constraint into the first term of (5), it would simplify to `$\text{tr}(\tilde{X}^\top C^\top\Theta C\tilde{X})=\text{tr}(X^\top\Theta X)$`, which would simplify the optimization significantly. However, instead of enforcing this constraint exactly, the paper simply introduces an error term `$|X-C\tilde{X}|$`, which seems unelegant to me. Why don't you enforce this constraint exactly?` \
> 	Note that we cannot just assume a constraint to be true and substitute its value accordingly; it is a constraint to be checked. Also it is very difficult to solve for this constraint exactly because it involves both $C$ and $\tilde{X}$ as variables, making it non-convex; that is why we add a relaxation of the constraint.
> 3. `You mention that the log determinant term can be written as the sum of the log of the eigenvalues, and that this ensures that a 'minimal' number of eigenvalues are zero. However, doesn't this ensure that not a single eigenvalue is zero?` \
> 	Yes, it tries to ensure that but any graph Laplacian will always have at least 1 zero eigenvalue (because there is always at least 1 connected component in any graph). So it is safe to add this term.
> 4. `Is the complexity that is described in the "Complexity analysis" paragraph the complexity of a single iteration or of all the iterations until convergence?` \
> 	We have described the complexity of one iteration/epoch, and compared it with other methods in Supplementary Material Section K.
> 5. `I see that you rescaled the NMI, ARI and modularity to percentages. This is okay for NMI (though I'm not a fan of it), but for ARI it is confusing because ARI can be negative. For modularity, I have no idea how this rescaling is done, because the upper bound of modularity is smaller than 1 (and incredibly expensive to compute).` \
> 	We have simply multiplied the values by 100; both modularity and ARI can be negative.
> 6. `Why do you draw lines in Figure 2b instead of making a table? The points that are connected don't correspond to consecutive things.` \
> 	The difference in the values and the impact is better understood as a graph. The points connected represent the same dataset. If we were to just use points (we tried that before), it would not look obvious which points belong to which dataset.
>
> We hope these answer your questions along with the PDF that will be updated soon and that you will consider raising your rating.

---

### Meta-Review · Area_Chair_izzF · 2023-12-05

**Metareview:**

The paper introduces a new algorithm for graph clustering based on the combination of modularity and graph coarsening. In particular, the paper introduces a new GNN architecture that captures together graph coarsening and modularity insights.

The paper contains some interesting ideas but it is not ready for publication. More specifically, several important concerns have been raised during the review period:

- the proposed method is not particularly novel

- the paper is a bit too dense and it is hard to read

- the experimental results are a bit unconvincing

Overall, the paper has some merits but it is below the ICLR acceptance bar

**Justification For Why Not Higher Score:**

The paper has some important limitations listed above and it is not ready for being published

**Justification For Why Not Lower Score:**

N / A

---

### Decision · Program_Chairs · 2024-01-16

Reject